# A role for ascorbate conjugates of (+)-catechin in proanthocyanidin polymerization

Keji Yu [1,2], Richard A. Dixon [3] & Changqing Duan [1,2✉]

Proanthocyanidins (PAs) are natural polymers of flavan-3-ols, commonly (+)-catechin and (−)-epicatechin. However, exactly how PA oligomerization proceeds is poorly understood. Here we show, both biochemically and genetically, that ascorbate (AsA) is an alternative "starter unit" to flavan-3-ol monomers for leucocyanidin-derived (+)-catechin subunit extension in the *Arabidopsis thaliana anthocyanidin synthase* (*ans*) mutant. These (catechin)$_n$:ascorbate conjugates (AsA-[C]$_n$) also accumulate throughout the phase of active PA biosynthesis in wild-type grape flowers, berry skins and seeds. In the presence of (−)-epicatechin, AsA-[C]$_n$ can further provide monomeric or oligomeric PA extension units for non-enzymatic polymerization in vitro, and their role in vivo is inferred from analysis of relative metabolite levels in both Arabidopsis and grape. Our findings advance the knowledge of (+)-catechin-type PA extension and indicate that PA oligomerization does not necessarily proceed by sequential addition of a single extension unit. AsA-[C]$_n$ defines a new type of PA intermediate which we term "sub-PAs".

[1] Center for Viticulture and Enology, College of Food Science & Nutritional Engineering, China Agricultural University, Beijing 100083, China. [2] Key Laboratory of Viticulture and Enology, Ministry of Agriculture and Rural Affairs, Beijing 100083, China. [3] BioDiscovery Institute and Department of Biological Sciences, University of North Texas, Denton, TX 76203, USA. ✉email: chqduan@cau.edu.cn

Proanthocyanidins (PAs) are oligomers and polymers of flavan-3-ols widely present in plants[1]. The content and composition of PAs from plant raw materials largely affect the bitterness and astringency of food products, including red wine and tea[2,3]. In addition, PAs possess beneficial properties against neurological and cardiovascular diseases in humans[4,5]. Understanding the routes of PA biosynthesis will help improve PA traits in commercial crops via genetic manipulation and viticulture practice.

The polymerization of PAs requires flavan-3-ol monomers as starter units and flavan-3-ol intermediates with an active C4 as extension units; these attack the C-8 or C-6 position of the starter unit and growing PA chain in a non-enzymatic manner[6-8]. (+)-Catechin and (−)-epicatechin are the two common PA subunits, with 2,3-trans and 2,3-cis stereochemistry respectively on their benzopyran rings. In the PA biosynthesis pathway, which can vary in a species-specific manner, dihydroflavonol reductase (DFR) first converts (+)-dihydroquercetin to reactive leucocyanidin[9], proposed for decades as the (+)-catechin-type extension unit[10]. Leucocyanidin is converted by leucocyanidin reductase (LAR) to (+)-catechin[11,12] in those species that possess this enzyme (Fig. 1). At the same time, a conserved anthocyanidin synthase (ANS) competes with LAR for leucocyanidin, channeling the flux to anthocyanidin reductase (ANR) to generate (−)-epicatechin-type PA subunits, although the product of ANS in plants can be either flav-3-en-3,4-diol or cyanidin[13-16] (Fig. 1). 4β-(S-Cysteinyl)-epicatechin (Cys-EC) can act as a (−)-epicatechin-type PA extension unit and serves as a second substrate of LAR to produce (−)-epicatechin monomer in Medicago truncatula[6] (Fig. 1). The absence of Cys-EC in M. truncatula anr mutant suggests that Cys-EC synthesis relies on ANR activity[6]. Leucoanthocyanidin dioxygenase (LDOX), a homolog of ANS, functions downstream of LAR in some species to convert (+)-catechin to flav-2-en-3-ol, another substrate for ANR to produce (−)-epicatechin[15,16] (Fig. 1). Precursor labeling studies in M. truncatula suggest that the MtLAR-MtLDOX-MtANR branch, when present, mainly provides (−)-epicatechin PA starter units, whereas the MtANS-MtANR pathway is responsible for (−)-epicatechin-type PA extension unit production[15] (Fig. 1). This explains why M. truncatula possesses LAR but mainly accumulates (−)-epicatechin-type PA building blocks[17].

Unlike M. truncatula, grape (Vitis vinifera) possesses significant levels of (+)-catechin-type PA subunits in berries[18]. We recently found that 4β-(S-cysteinyl)-catechin (Cys-C) co-exists with its 2,3-cis isomer Cys-EC in grape berries and confirmed that Cys-C is another (+)-catechin-type PA extension unit[11,15] (Fig. 1). The knock-out of LAR in M. truncatula does not affect the accumulation of Cys-C in the pods, suggesting that Cys-C biosynthesis is LAR-independent[11]. In vitro studies showed that Cys-C is generated non-enzymatically by co-incubation of leucocyanidin and Cys, suggesting that the (+)-catechin moiety of Cys-C is sourced from leucocyanidin[16] (Fig. 1). Arabidopsis thaliana does not possess LAR and LDOX homologs and exclusively produces (−)-epicatechin-based PAs[19]. This means that the ANS-ANR pathway contributes to the synthesis of both (−)-epicatechin-type PA starter and extension units in this species (Fig. 1). In the A. thaliana ans mutant, essentially reflecting the absence of both ANS and LAR, leucocyanidin and Cys-C are detected; these are not present in wild-type plants[8,16,20]. However, although the A. thaliana ans mutant possesses (+)-catechin-type PA extension units (leucocyanidin and Cys-C), this mutant is PA-deficient[13]. Similarly, in the corresponding M. truncatula lar:ans double mutant, PA starter units and oligomers were not detectable, even though (+)-catechin-type PA extension units were released by phloroglucinolysis[15]. How plants cope with (+)-catechin-type extension units in the absence of PA starter units, therefore, remains unclear.

While investigating the destination of (+)-catechin-type PA extension units in the A. thaliana ans mutant, we have identified ascorbic acid (AsA) as an alternative to flavan-3-ol monomers for trapping leucocyanidin-derived PA extension units, resulting in AsA conjugates of monomeric and oligomeric (+)-catechin (AsA-[C]$_n$). These conjugates are also found in wild-type grape berries during the active stages of PA biosynthesis. In vitro studies demonstrated that these AsA conjugates (here termed "sub-PAs") further function as extension units for non-enzymatic PA polymerization in the presence of flavan-3-ol starter unit. These results provide a further understanding of mechanisms for stabilization of reactive PA extension units and indicate that PA polymerization is not restricted to the addition of single monomeric units to the growing chain.

## Results

**PA composition in the A. thaliana ans mutant**. The A. thaliana ans null mutant (ans-4) exhibits a strong PA-deficient phenotype with a transparent testa and weak p-dimethylaminocinnamaldehyde (DMACA) staining, but the residual PAs contain (+)-catechin-type extension units (leucocyanidin and Cys-C)[8,13,20]. To better characterize the PA composition in the ans-4 mutant, soluble PA fractions of siliques at 7 days-after-pollination (DAP) were isolated with 70% (v/v) acetone followed by chloroform extraction to remove chlorophyll. High-performance liquid chromatography-triple quadrupole mass spectrometry (HPLC-QqQ) analysis showed that ans-4 produced only trace levels of (−)-epicatechin and nearly no PA dimer and trimer by reference to standards and corresponding extracts from wild-type (WT) siliques (Fig. 2a–c). Loss of (−)-epicatechin-based PAs is consistent with the impaired ANS activity, blocking the biosynthesis of (−)-epicatechin-type PA starter and extension units.

Proanthocyanidins are converted to colored anthocyanidins under hydrolysis in acidic butanol[1,21]. To visualize and quantify PA extension units, we applied butanolysis to soluble PAs, insoluble PAs (in residues after acetone/water extraction) and total PAs (in samples with chlorophyll removed but without acetone/water extraction) from both ans-4 and WT 7 DAP siliques. All ans-4 samples gave a red color, although less intense than that of the corresponding WT samples (Fig. 2d). This is consistent with the mass spectrometry (MS) evidence for the existence of (+)-catechin-type PA extension units in ans-4[8]. Further quantification of total PA hydrolysis products at A$_{550}$ suggested that the total amount of extension units in ans-4 mutant reached 75% of that in the WT (Fig. 2e). The nature of such large amounts of extension units in the ans-4 mutant is puzzling.

Leucocyanidin is converted to colored cyanidin under butanolysis[22]. To determine to what extent free leucocyanidin contributes to the colored products in acid-butanol treated extracts of the ans-4 mutant, we measured leucocyanidin levels in 7 DAP siliques of ans-4 using ultra-high-performance liquid chromatography-quadrupole time-of-flight mass spectrometry (UHPLC-QToF) (Supplementary Fig. 1). The concentration of free leucocyanidin was ~1500-fold less than that of total PA extension units quantified by butanolysis, indicating that leucocyanidin produced in the ans-4 mutant is largely consumed or polymerized. Because the Cys-C standard also gave a red color after butanolysis (Supplementary Fig. 2), it is hard to tell whether the known (+)-catechin-type extension units in ans-4 were truly coupled into PAs. However, given that high molecular weight PAs are insoluble[6], the red color of the insoluble PA butanolysis products suggests that (+)-catechin-type extension units might be polymerized in ans-4 (Fig. 2d). These results raise the possibility that there might be an alternative compound to flavan-3-ol monomer as the starter for PA extension in this mutant.

**Fig. 1 Alternative pathways to PA starter and extension units.** (−)-Epicatechin-type PA starter and extension unit biosynthesis pathways in Medicago (*M. truncatula*) occur through separate by ANS-ANR and LAR-LDOX-ANR branches[6,15,16], while (−)-epicatechin-type PA starter and extension unit biosynthesis in Arabidopsis (*A. thaliana*) and grape (*V. vinifera*) are all derived from the ANS-ANR pathway[11,13,14]. In grape, (+)-catechin-type PA starter unit is synthesized by LAR and (+)-catechin-type PA extension units are derived from leucocyanidin[11]. DFR, dihydroflavonol reductase; ANS, anthocyanidin synthase; LDOX, leucoanthocyanidin dioxygenase; LAR, leucoanthocyanidin reductase; ANR, anthocyanidin reductase. CoA, coenzyme A; Cys-C, 4β-(*S*-cysteinyl)-(+)-catechin; Cys-EC, 4β-(*S*-cysteinyl)-(−)-epicatechin.

**Screening for compounds with a (+)-catechin backbone in the *A. thaliana ans* mutant.** MS/MS screening failed to detect the presence of Cys conjugates of (+)-catechin (Cys-C-C or Cys-C-C-C) in the *ans-4* mutant, ruling out the possibility for Cys-C as the nucleophilic acceptor for (+)-catechin-type PA extension units. A hypothetical structure was therefore proposed (Fig. 3a) in which the unknown starter unit (denoted as "X") was conjugated with C4 of (+)-catechin. Theoretically, the (+)-catechin backbone of "X" would be detected in the forms of carbocation (*m/z* 289) or the corresponding quinone (*m/z* 287) under a negative mode MS ion source. If "X" could be also ionized, the *m/z* change from the precursor ion would be equivalent to a loss of mass of 288 (Fig. 3a). According to these deductions, a neutral loss scan of 288 (precursor *m/z* range from 300 to 1000) was applied to

explore the compounds with (+)-catechin backbone in the extracts of 7 DAP siliques from the *A. thaliana ans-4* mutant using HPLC-QqQ. Two procyanidin dimer standards were scanned in parallel and could be successfully identified, indicating that the screening method was reliable (Fig. 3b). Compared with WT, three compounds unique to *ans-4* were detected (Fig. 3b) and the *m/z* values of their precursor ions were 625, 463 and 465 respectively (Fig. 3c). Only the compound with *m/z* 463 possessed the MS2 product ion at *m/z* 289, an indicator of a (+)-catechin group (Fig. 3d).

The extracts of *ans-4* and WT siliques were re-analyzed by UHPLC-QToF to obtain higher resolution MS information for structure deduction. By extracting ions from 462.8000 ± 20 ppm to 463.2000 ± 20 ppm, the candidate was found to have a

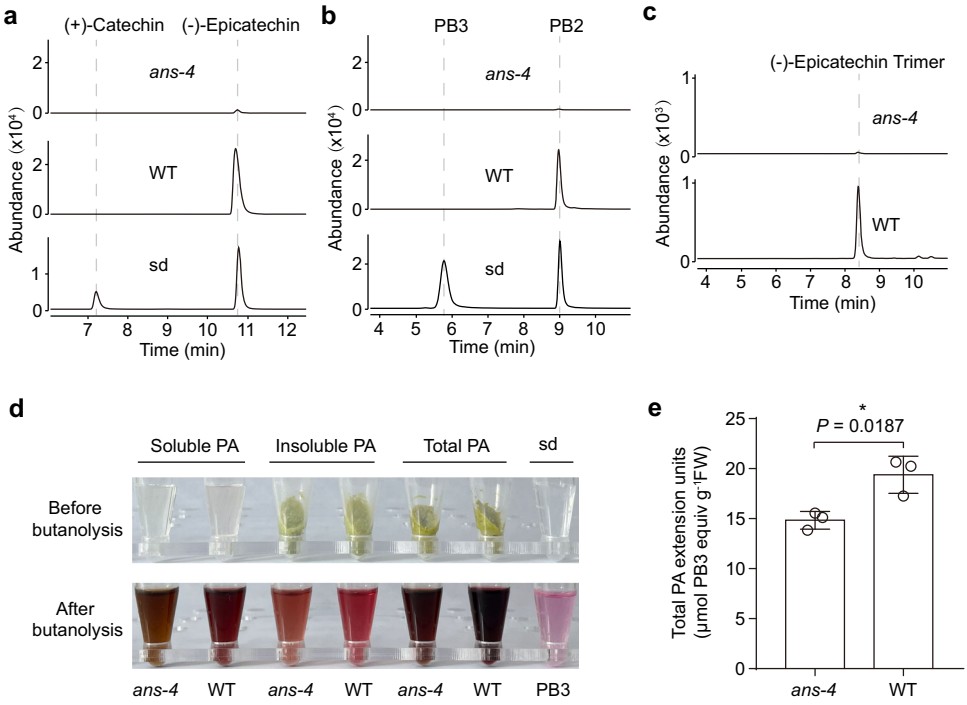

**Fig. 2 PA composition analysis in 7 DAP siliques of the *A. thaliana ans-4* mutant. a–c** HPLC-QqQ chromatograms for characterization of the soluble PA fraction in 7 DAP siliques of the *A. thaliana ans-4* mutant and wild-type (WT) plants. **a** MRM transition of *m/z* (289 → 123) showing that trace amounts of (−)-epicatechin remained in the *ans-4* mutant. **b** MRM transition of *m/z* (577 → 407) showing that the *ans-4* mutant has lost B-type procyanidin dimer. **c** MRM transition of *m/z* (865 → 407) showing that the *ans-4* mutant has lost B-type procyanidin trimer. **d** Visualization of the presence of PA extension units in 7 DAP siliques of *ans-4* mutant and WT by the butanolysis method. The upper panel shows the non-lyophilized soluble PA and the residues containing insoluble PA and total PA before applying butanolysis. Soluble PA was extracted with acetone/water followed by chloroform extraction to remove chlorophyll. Insoluble PA remained in the residues after soluble PA extraction. The residues containing total PA were obtained as the ground tissue only with chloroform extraction to remove chlorophyll. The lower panel shows the butanolysis supernatant of the lyophilized soluble PA and the residues containing insoluble PA and total PA after reaction. Procyanidin B3 (0.25 mM) standard (sd) served as the positive control. **e** Quantification of total PA extension units in 7 DAP siliques of *ans-4* mutant and WT. PA extension levels were determined by the butanol-HCl method and expressed as procyanidin B3 (PB3) equivalents. FW: fresh weight. Data are shown as the mean ± SD (for *n* = 3 biologically independent samples; *$P < 0.05$, two-tailed unpaired Student's *t* tests). Source data of Fig. 2e are provided as a Source Data file.

retention time at 5.066 min on UHPLC with an *m/z* of 463.0863 ± 20 ppm on QToF (Supplementary Fig. 3). The MS2 profile of the candidate was further obtained by targeted MS/MS analysis (Fig. 4a). The ion fragment characteristics of the candidate compound were similar to that of (−)-epicatechin-5-*O*-glucuronide earlier identified in *M. truncatula*[6]. However, *ans-4* mutant lacks the ability to synthesize flavan-3-ol monomers, the substrate for further glucuronidation. Furthermore, the absence of a specific ion at *m/z* 113 in the MS2 profile excluded the possibility that the fragment *m/z* 175 was a glucuronic acid moiety[23]. Thus, the product ion at *m/z* 175.0251 ± 20 ppm was annotated as ascorbate (AsA) with the theoretical *m/z* 175.0248, which was further supported by the presence of a specific ion at *m/z* 115.0031 ± 20 ppm and one rearranged from the ethane-1,2-diol moiety at *m/z* 59.0139 ± 20 ppm (Fig. 4a). The fragment at *m/z* 289.0718 ± 20 ppm might come from the deprotonated (+)-catechin backbone. However, it was also possible that this could represent deprotonated (−)-epicatechin, since ANR might contribute epimerase activity in the *ans-4* mutant[24]. AsA is present predominantly as the 3-OH deprotonated anion at pH 7 and possesses nucleophilic reactivity[25], suggesting that AsA could attack flavan-3-ol carbocations at neutral pH to form AsA conjugates of (+)-catechin or (−)-epicatechin. Therefore, the candidate compound was likely to be either 4-(*O*-ascorbate)-(+)-catechin (AsA-C) or 4-(*O*-ascorbate)-(−)-epicatechin (AsA-EC), with theoretical *m/z* of 463.0882 (Fig. 4a).

To verify the flavan-3-ol backbone, AsA was incubated with (+)-catechin carbocation donor (Cys-C or leucocyanidin)[10,11] or (−)-epicatechin carbocation donor (Cys-EC)[6] at pH 7.4 for 1 h (Fig. 4b). The resulting solutions were analyzed by UHPLC-QToF along with the extracts from *ans-4* siliques. Each yielded one compound at *m/z* 463.0882 ± 20 ppm with the same MS2 profile as the candidate in the *ans-4* sample (Fig. 4c, d). The candidate in *ans-4* and the reactions containing Cys-C or leucocyanidin possessed the same retention time on UHPLC from 4.994 to 5.217 min, ahead of the retention time from 6.168 to 6.370 min of the product of incubation of AsA with Cys-EC (Fig. 4c). As a result, the compound of interest in *ans-4* was identified as AsA-C.

**AsA is an alternative starter unit for (+)-catechin-type PA extension in planta.** To investigate whether AsA-C could provide a basis for the further polymerization of flavan-3-ol blocks, we performed MS/MS analysis of the *ans-4* extract using the theoretical *m/z* of AsA-C-C (751.1615) and AsA-C-C-C (1039.2150) as targets. WT extract and the reaction containing AsA and leucocyanidin were analyzed in parallel as references. In both the *ans-4* extract and the chemical reaction, a precursor at *m/z* 751.1615 ± 20 ppm was detected in MS1 at 4.01 min and the corresponding MS2 profile between the two samples was the same (Fig. 5a, b). The product ion with *m/z* at 175.0251 ± 20 ppm corresponded to the dissociated AsA moiety (Fig. 5b and

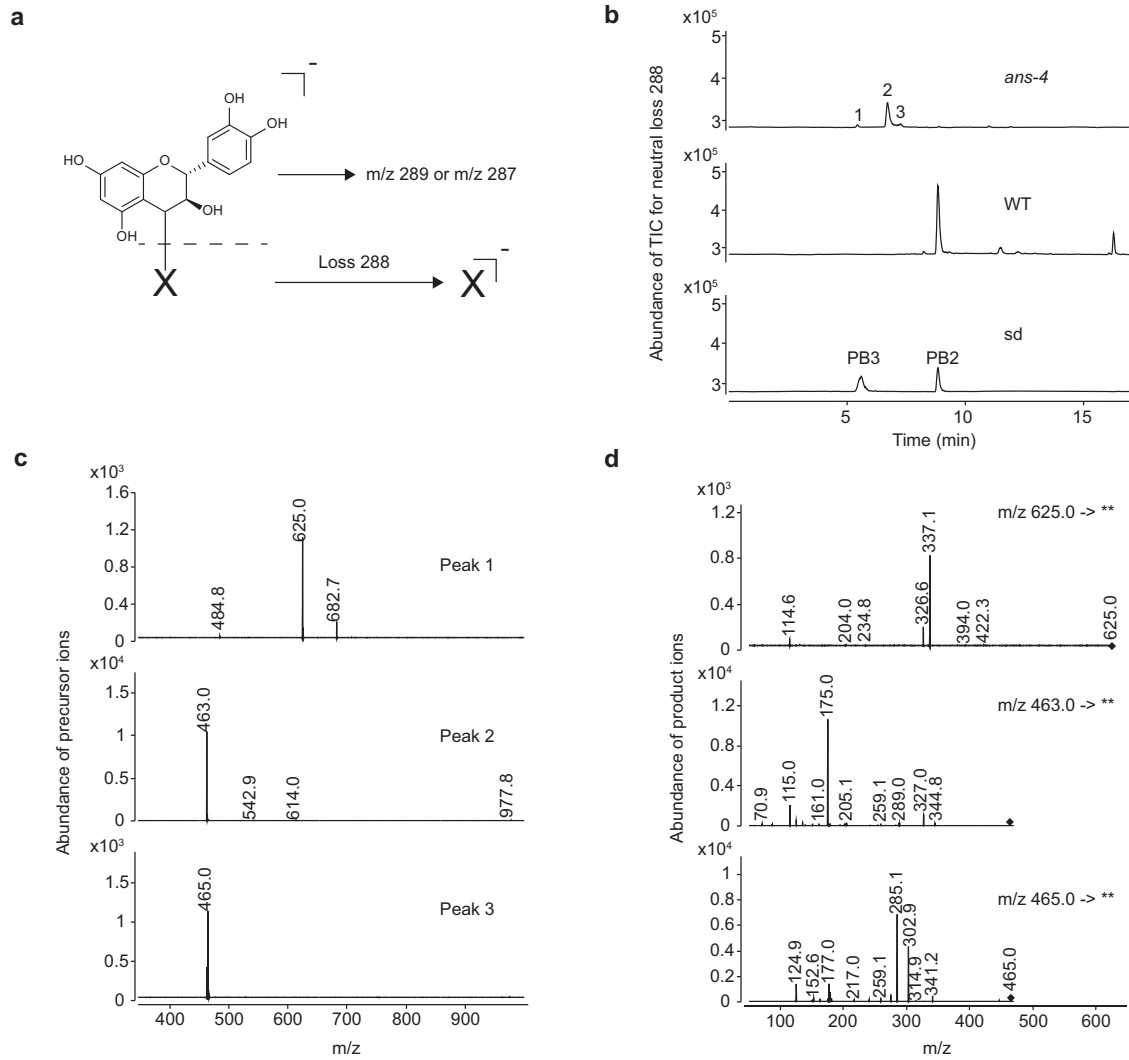

**Fig. 3 Screening of compounds with a (+)-catechin backbone in the *A. thaliana ans-4* mutant. a** Ion fragmentation schemes for the (+)-catechin-unknown compound "X" adduct in mass spectra. **b** Neutral loss scan at mass 288 for the soluble PA fraction from 7 DAP siliques of *A. thaliana ans-4* mutant and wild-type (WT) plants. Procyanidin B3 (PB3) and procyanidin B2 (PB2) served as positive controls. **c** *m/z* of candidate compounds shown in (**b**). **d** Product ion profiles of the compounds at *m/z* 625, *m/z* 463 and *m/z* 465 obtained on HPLC-QqQ.

Supplementary Fig. 4), and the fragments with *m/z* at 287.0549 ± 20 ppm and 575.1149 ± 20 ppm were the specific ions resulting from the quinone methide (QM) fission after the cleavage of the C4-bond of the C-ring or F-ring of procyanidin oligomers[26] (Supplementary Fig. 4). The ion with *m/z* at 449.0843 ± 20 ppm was derived from heterocyclic ring fission (HRF) of the C-ring of the *m/z* 575.1195 ion (Supplementary Fig. 4), which was connected to the D-ring with an interflavan linkage (IFL) at C4-C8 or C4-C6[27]. Previous in vitro studies suggested that the incorporation of leucocyanidin into PA polymers resulted from the addition of exclusively (+)-catechin-type upper units to mainly form IFLs with the lower unit at C4-C8 and to a lesser degree at C4-C6[10,28]. Our data confirm a trapped leucocyanidin-derived compound in *ans-4* extracts as AsA-C-C, with an IFL between (+)-catechin units of the C4-C8 type. A compound with the same MS1 peaks as AsA-C-C was found in the WT sample at around 6 min on UHPLC (Fig. 5a). However, MS2 data excluded the possibility of this being an isomer of AsA-C-C due to the absence of the AsA-specific ion at *m/z* 175.0248 (Fig. 5b).

The extraction of the MS1 ion with theoretical *m/z* of AsA-C-C (1039.2150 ± 20 ppm) gave two candidates at 5.03 and 5.71 min in both the *ans-4* and the in vitro synthesis mixture, without the corresponding signals in the WT extract (Fig. 5c). The MS2 profiles of the four precursor ions were the same, including the four characteristic ions of AsA-C-C (Fig. 5d). The product ion at *m/z* 863.1829 ± 20 ppm was consistent with a QM fission at the I-ring of a procyanidin trimer moiety (Fig. 5d, e), indicating that the precursor possessed one more (+)-catechin unit than AsA-C-C (Supplementary Fig. 4). Furthermore, the fragment at *m/z* 737.1512 ± 20 ppm suggested that the newly added catechin unit was linked to the lower unit via a C4-C8 or C4-C6 IFL (Fig. 5e and Supplementary Fig. 5). Among the two products from the reaction between leucocyanidin and AsA, the compound appearing at 5.03 min was more abundant than the one emerging at 5.71 min. Based on previous work[10,28], the compound with higher abundance should exclusively contain C4-C8 IFLs between (+)-catechin building blocks (Fig. 5e), whereas the much less abundant compound is likely to possess a C4-C6 IFL after addition of one (+)-catechin unit to AsA-C-C (Supplementary Fig. 5). Thus, the compounds at 5.03 min and 5.71 min in *ans-4* extracts were annotated as AsA-C-C-C (AsA-catechin-[4,8:4,8]-bi-catechin) and its isomer AsA-catechin-[4,8:4,6]-bi-catechin.

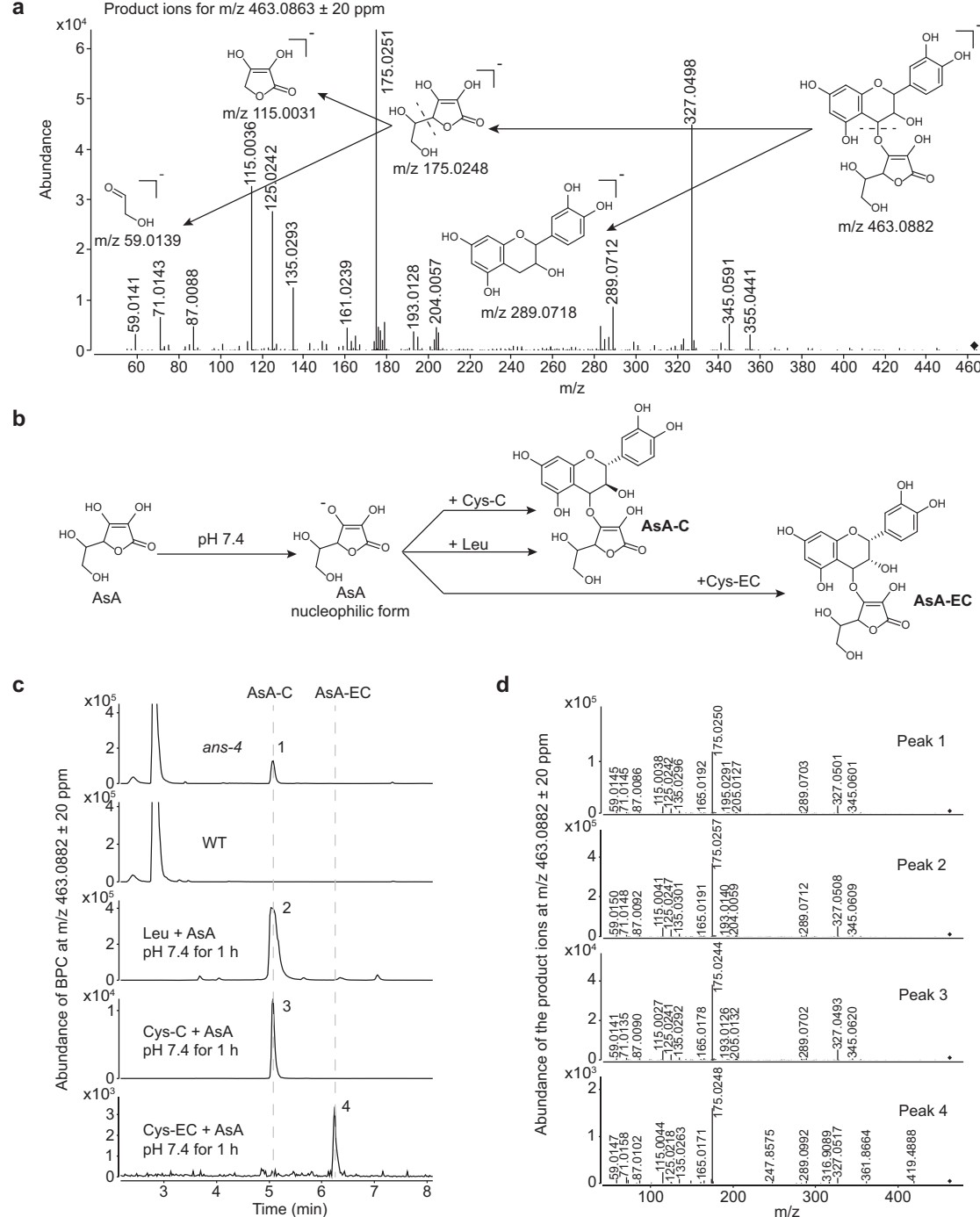

**Fig. 4 The candidate compound is AsA-C. a** Product ions for fragmentation of the catechin conjugate of *m/z* at 463.0863 ± 20 ppm and deduction of its fragmentation mode. Theoretically calculated *m/z* values are labeled under the corresponding structures. **b** Scheme for the chemical synthesis of AsA-C and AsA-EC in vitro. **c** Comparison of the retention time of the precursor ions at *m/z* 463.0882 ± 20 ppm between the *A. thaliana ans-4* extract and the product of the chemical reaction. **d** Product ions of compounds marked in (**c**).

In the same way, we detected two compounds matching the MS/MS profile of AsA-C-C-C-C (pentamer) in the *ans-4* mutant (Supplementary Fig. 6a, b). However, the instrument sensitivity for capturing the pentamer precursor ion was too low (Supplementary Fig. 6a), reflecting the common challenge of characterizing typical PA polymers by LC/MS[29,30]. As AsA-[C]$_n$ could yield a series of fragments $[(C_{15}H_{12}O_6)_m - H]^-$ (0 < m ≤ n) resulting from QM fission during the IFL cleavage or the leaving of the AsA moiety (Supplementary Fig. 6b), we re-analyzed soluble PAs from *ans-4*, together with WT samples and products of AsA + Leu incubation,

by ultra-high performance liquid chromatography-triple quadrupole mass spectrometry (UHPLC-QqQ) with multiple reaction monitoring (MRM) of the transition of [M-H]⁻ *m/z* 1327.3 to $[(C_{15}H_{12}O_6)_4 - H]^-$ *m/z* 1151.2. Two peaks representing pentamers were clearly detected in the *ans-4* and AsA + Leu samples but not in the WT (Supplementary Fig. 6c), suggesting that the MRM approach could improve the sensitivity for detecting AsA conjugates with higher molecular weight. We then used MRM transitions of *m/z* (1615.3 → 1151.2), *m/z* (1904.4 → 1151.2), and *m/z* (2192.5 → 1727.4) with optimized MS parameters to analyze

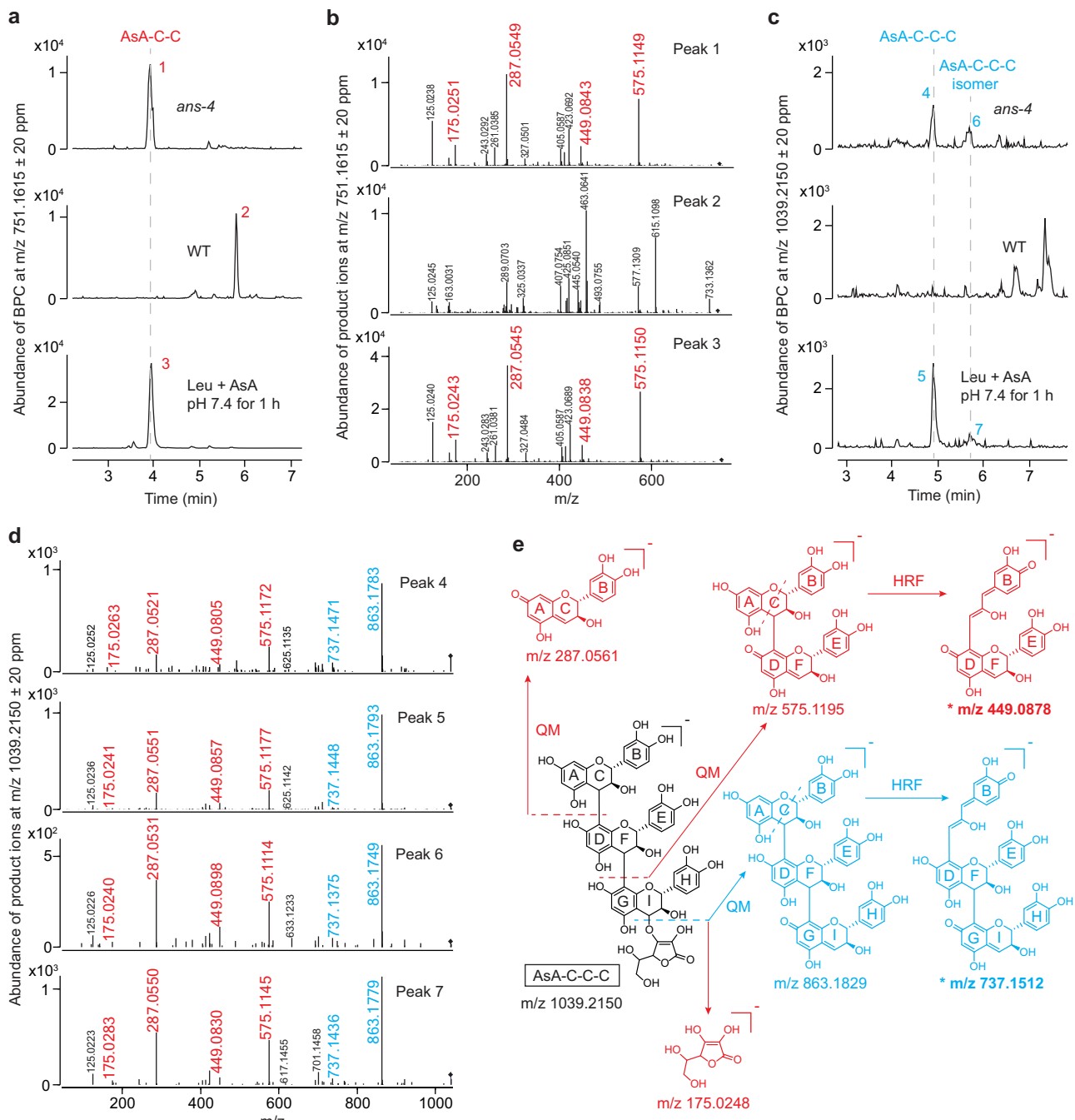

**Fig. 5 AsA-C-C and AsA-C-C-C accumulate in 7 DAP siliques of the *A. thaliana ans-4* mutant.** The soluble PA fraction of 7 DAP siliques of the *A. thaliana ans-4* mutant was analyzed in parallel with the product of reaction of AsA + leucocyanidin (Leu) as the standard using UHPLC-QToF. **a** Base peak chromatogram (BPC) showing that a compound with the same *m/z* (751.1615 ± 20 ppm) as AsA-C-C accumulates in the *A. thaliana ans-4* mutant. **b** Product ions of the compounds marked in (**a**). The fragmentation mode deduction supporting the conclusion that compounds 1 and 3 are AsA-C-C is shown in Supplementary Fig. 4. **c** BPC showing that the compounds with the same *m/z* (1039.2150 ± 20 ppm) as AsA-C-C-C accumulate in the *A. thaliana ans-4* mutant. **d** Product ions of compounds marked in (**c**). **e** The fragmentation mode deduction supporting the conclusion that compounds 4 and 5 in (**c**) are AsA-C-C-C based on the product ion profiles. The asterisks indicate the critical fragments supporting the natural IFL. The corresponding scheme supporting the conclusion that compounds 6 and 7 are AsA-C-C-C isomers is shown in Supplementary Fig. 5. The *m/z* values and the chemical formulae in red are shared by both AsA-C-C and AsA-C-C-C, whereas the *m/z* values and the formulae in blue are unique to AsA-C-C-C.

the above samples. Each analysis trapped one compound that existed in both the *ans-4* and AsA + Leu system but not in the WT, confirming the presence of hexamer, heptamer, and octamer, respectively (Supplementary Fig. 6d, e, f).

To provide further insights into the average chain length of soluble AsA-[C]$_n$ in the *ans-4* mutant, we carefully evaluated the

standard method for measuring the mean degree of polymerization (mDP) of PAs. Conventionally, this requires acid-catalysis to cleave the IFL in the presence of excess nucleophile, producing flavan-3-ol-nucleophile adducts (extension units) and flavan-3-ol monomers (starter units). A pilot study suggested that the widely used phloroglucinolysis was not suitable in this case, as

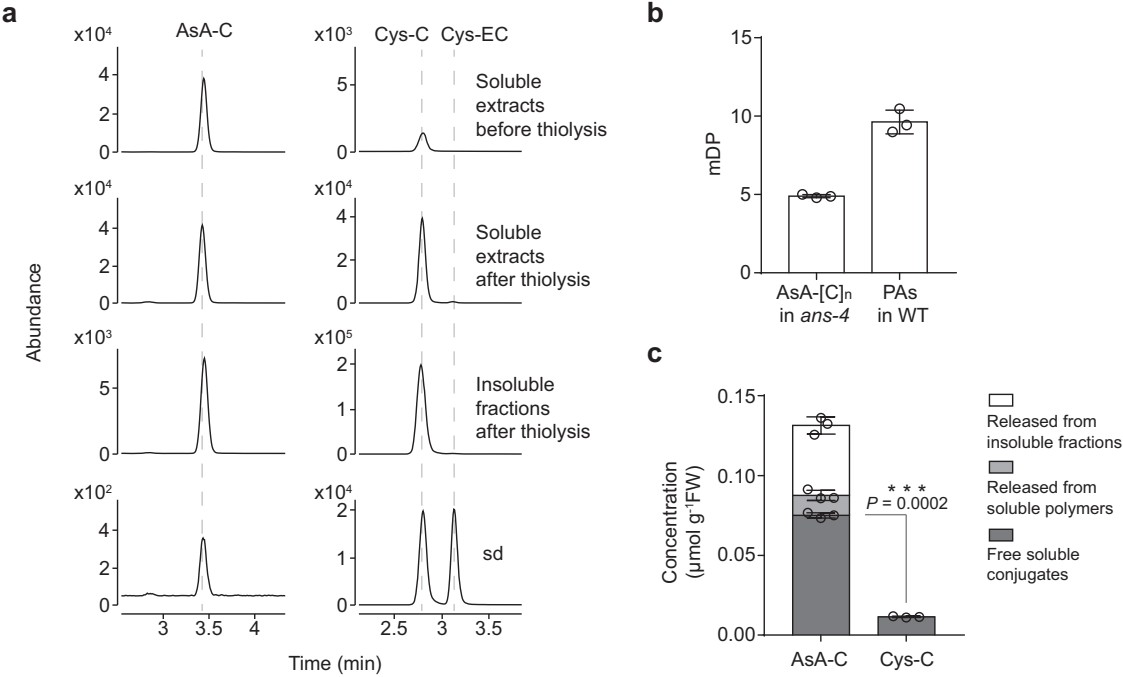

**Fig. 6 Thiolysis analysis of the soluble extracts and the insoluble fractions of 7 DAP siliques of *ans-4* mutant in the presence of excess Cys. a** UHPLC-QqQ analysis of AsA-C with MRM transition of $m/z$ (463.1 → 175.0), and Cys-C and Cys-EC with MRM transition of $m/z$ (408.1 → 125.0) in the soluble extracts before and after thiolysis and the insoluble fractions after thiolysis in the presence excess Cys. sd: standard. Note that the ion abundances are not directly comparable among different types of samples, as different dilution factors were applied to each sample to prevent the over-saturation of ion abundance in the QqQ detector. **b** The mDP of soluble AsA-[C]$_n$ in the *ans-4* mutant and soluble PA in WT. Data are shown as the mean ± SD (for $n = 3$ biologically independent samples). **c** The concentration of AsA-C from different fractions and free Cys-C in *ans-4* mutant sample. FW: fresh weight. Data are shown as the mean ± SD (for $n = 3$ biologically independent samples; ***$P < 0.001$, two-tailed unpaired Student's $t$ tests). Source data of Fig. 6b and c are provided as a Source Data file.

phloroglucinol could partially substitute for the AsA and Cys moieties of chemically synthesized AsA-C and Cys-C standards, rendering the extension units in AsA-[C]$_n$ and (+)-catechin backbone from Cys-C indistinguishable (Supplementary Fig. 7). In contrast, after performing thiolysis with excess Cys, the AsA-C and Cys-C standards remained at the same level as before the reaction (Supplementary Fig. 8), indicating that this IFL cleavage method had no effect on AsA-C and Cys-C. Under this treatment, the first two subunits of AsA-[C]$_n$ will be detected as AsA-C, and the content of the second and the subsequent extension units can be calculated by subtracting the Cys-C before lysis from that after lysis. The products before and after thiolysis from *ans-4* and WT samples were detected using MRM on a UHPLC-QqQ and quantified from the curves developed with the corresponding standards (Fig. 6a and Supplementary Fig. 9). The mDP of soluble PAs in WT samples was determined from Eq. (1) (Methods section) to be around 10 (Fig. 6b), consistent with the value of 9.4 recently reported using phloroglucinolysis[15]. The mDP of soluble AsA-[C]$_n$ in the *ans-4* mutant was calculated as 5 from Eq. (2) (Methods section), consistent with the existence of dimer to octamer in these AsA conjugates (Fig. 6b). Then, residues from *ans-4* after soluble PA extraction were subjected to thiolysis with Cys, and AsA-C and Cys-C were both detected in the lysis products using UHPLC-QqQ (Fig. 6a), indicating that this mutant line possesses large AsA-[C]$_n$ polymers that are insoluble. We also observed that the molar concentration of AsA-C was already seven times higher than that of Cys-C in the soluble PA fraction of *ans-4* before thiolysis (Fig. 6c), suggesting that AsA could capture (+)-catechin-type PA extension units more efficiently than Cys in vivo. Quantification of AsA-C released from the soluble oligomers and insoluble fractions after

thiolysis showed that more than 40% of the AsA-C produced in the *ans-4* mutant was derived from larger polymers (DP > 2) (Fig. 6c).

AsA-C, AsA-C-C and AsA-C-C-C (AsA-[C]$_n$) could also be detected in extracts from 7 DAP siliques of the weak *ans-2* allele isolated by Abrahams et al.[13], although their abundance on MS1 was much lower than in extracts of the *ans-4* null mutant (Supplementary Fig. 10). This suggests that the biosynthesis of AsA-[C]$_n$ in planta is dependent on the accumulation of leucocyanidin due to the loss of ANS activity.

To gain insights into the relationship between AsA level and AsA-[C]$_n$ accumulation, we crossed *ans-4* with the AsA-deficient mutant *vtc2-5*, which is reported to possess ~70% less total cellular AsA than the WT without decreased growth[31]. No developmental difference was observed between *vtc2-5:ans-4* and *ans-4* mutants. Compared with *ans-4*, AsA-[C]$_n$ in the soluble extracts of *vtc2-5:ans-4* 7 DAP siliques were largely decreased, while Cys-C was at the same level (Supplementary Fig. 11). This indicates that AsA-[C]$_n$ level is closely associated with AsA availability. In the 7 DAP siliques of the *vtc2-5* single mutant, levels of both soluble and insoluble PAs were slightly lower than those of WT (Supplementary Fig. 12), consistent with the involvement of AsA as a co-substrate for flavanone 3-hydroxylase (F3H), the enzyme upstream of DFR in flavonoid biosynthesis[32]. AsA deficiency should limit leucocyanidin synthesis in *vtc2-5:ans-4*. To test this, thiolysis with excess Cys was performed on both soluble extracts and insoluble fractions of the 7 DAP siliques from the two mutants in order to trap (+)-carbocations derived from leucocyanidin and released from the cleavage of IFLs of AsA-[C]$_n$ under strong acid conditions. The amount of AsA-C and Cys-C in the lysis products was used

to compare the total leucocyanidin level between *vtc2-5:ans-4* and *ans-4* mutants. Levels of AsA-C and Cys-C were lower in soluble extracts of *vtc2-5:ans-4* than of *ans-4* after thiolysis, whereas the levels of the two compounds released from the insoluble residues were not different between the double and single mutants (Supplementary Fig. 13). Thus, the low content of AsA could restrict leucocyanidin production in the *vtc2-5:ans-4* mutant, resulting in less stabilization of reactive leucocyanidin as AsA-$[C]_n$.

**AsA-$[C]_n$ exists in flowers, berry skins and seeds of grapevine at active stages of PA biosynthesis.** Cys-C exists in grape berries and functions as a (+)-catechin-type extension unit[11]. The genetic and in vitro evidence above suggests that AsA-$[C]_n$ and Cys-C share the same source of (+)-catechin backbone from leucocyanidin[8,16] (Fig. 4 and Supplementary Fig. 10). Thus, we speculated that AsA-$[C]_n$ should also exist in grapevine. To confirm this, grape tissues were sampled at different E-L development stages[33], as indicated in Fig. 7a, for soluble PA extraction followed by UHPLC-QToF analysis. AsA-C, AsA-C-C and AsA-C-C-C were mainly present in flowers, whole setting berries and skins and seeds of pea-sized berries, whereas the abundance of these compounds in berry skins and seeds dramatically dropped before verasion such that they were undetectable at harvest (Fig. 7b–i). PA pathway genes are expressed throughout early flowering and the development of berry skins and seeds until verasion[34]. Thus, the presence of AsA-$[C]_n$ in flowers and berries of grapes corresponds with the active PA biosynthesis period. As grape possesses a complete PA pathway in contrast to the *A. thaliana ans* mutant, these results support a role for AsA-$[C]_n$ in PA biosynthesis in wild-type plants.

**AsA-$[C]_n$ is a buffer pool of extension unit for non-enzymatic PA polymerization.** The AsA moiety of AsA-$[C]_n$ is linked at the C4 position of (+)-catechin, with the oxygen at the same position as an isovalent sulfur in Cys-C. The function of Cys-C as a PA extension unit depends on the auto-cleavage of the C4-S bond at neutral pH to form a (+)-catechin carbocation intermediate, which can attack the nucleophilic C8 of flavan-3-ol monomer or the last PA unit to extend the PA chain[6,11,15]. We estimated the bond dissociation energies (BDEs) within the known (+)-catechin conjugates using ALFABET[35], and the BDE of the C4-O bond in AsA-$[C]_n$ was predicted as 20% less than that of the C4-S bond of Cys-C (Supplementary Table 1). This means that the C4-O bond in AsA-$[C]_n$ is weaker than the C4-S bond of Cys-C and therefore more prone to dissociation. Thus, it is possible that AsA-$[C]_n$ might also release monomeric or oligomeric (+)-catechin-type carbocation intermediates for PA polymerization at neutral pH. To test this hypothesis, equimolar (100 μM) amounts of (−)-epicatechin and AsA-C or AsA-C-C were incubated at pH 7.4 for 1 h. The reaction between (−)-epicatechin and Cys-C was used to yield procyanidin oligomer standards with (−)-epicatechin as the starter unit and (+)-catechin as the extension unit[15]. UHPLC-QToF analysis showed that procyanidin dimer, trimer, tetramer, and pentamer were all formed when (−)-epicatechin was co-incubated with AsA-C or AsA-C-C, whereas there was no oligomeric procyanidin formed in the controls containing (−)-epicatechin, AsA-C or AsA-C-C as single substrates. (Fig. 8a–d). The amounts of trimer and pentamer (with uneven number DP) produced by the reaction between (−)-epicatechin + AsA-C-C were 15 and 20 times more than the amounts from the reaction between (−)-epicatechin + AsA-C, whereas the amounts of PA dimer and tetramer (with even number DP) from the former reaction were both around only 10% of that from the latter reaction (Fig. 8e). Theoretically, the

reaction of (−)-epicatechin with AsA-C can generate PAs with even or uneven DP, as each AsA-C can only release one (+)-catechin carbocation, thereby increasing the DP by 1 unit (Fig. 8f). The preference for the formation of uneven number DP PAs on incubating (−)-epicatechin with AsA-C-C reveals that AsA-C-C mainly provides (+)-catechin-dimer extension units, thereby increasing the DP by 2 units (Fig. 8f). A recent in vitro study of proanthocyanidin degradation behavior demonstrated that B-type procyanidin dimers undergo depolymerization due to oxidation at neutral pH[36]. Thus, the even number DP procyanidins present in incubating (−)-epicatechin with AsA-C-C are likely derived from the degradation of procyanidins with uneven number DP. Taken together, both AsA-C and AsA-C-C can function as the source of PA extension units. To extrapolate, it is possible that AsA-$[C]_n$ can provide from one to n extension units in a single PA polymerization event.

To address the relationship between (+)-catechin conjugates and the progress of (+)-catechin-type PA polymerization in planta, we measured the levels of AsA-$[C]_n$, Cys-C, and (+)-catechin extension unit in PAs in grape berry skins and seeds at the stages E-L 31 (pea-sized berry), E-L 33 (before version), E-L 35 (onset of verasion), E-L 35.5 (50% verasion) and E-L 36 (100% verasion), covering both active and inactive periods of PA biosynthesis. The soluble extracts were directly used for quantifying free (+)-catechin conjugates. The soluble PAs were further purified with Sephadex LH-20 resin following the published protocol[16], and neither Cys-C nor AsA-$[C]_n$ was detected in the purified products (Supplementary Fig. 14). Thus, Cys-C product formed after thiolysis with an excess of Cys could represent (+)-catechin extension unit integrated into soluble and insoluble PAs. In both skins and seeds, the overall level of AsA-$[C]_n$ continuously decreased from E-L 31 to E-L 35 and then remained at trace level or became undetectable (Fig. 9). No AsA-C was detected from the lysis products of insoluble grape PAs (Supplementary Fig. 15). Therefore AsA-$[C]_n$ in grape is not insolubilized as in the *A. thaliana ans* mutant (Fig. 6a) but rather participates in further reactions. Although free Cys-C content was highest at E-L 31 in both berry skins and seeds, no significant change was observed after a decrease at E-L 33. The accumulation patterns of the (+)-catechin extension unit in PAs were similar in grape skins and seeds. Before E-L 35.5, the amount of (+)-catechin extension unit in soluble PAs kept increasing, whereas that in the insoluble PAs remained unchanged. From E-L 35.5 to E-L 36, (+)-catechin extension unit level gradually decreased in soluble PAs but increased in insoluble PAs, consistent with reports that large amounts of soluble PAs become insolubilized post-verasion in PA-cell wall complexes[37]. Thus, at the active stages of PA synthesis in berries (E-L 31, E-L 33, and E-L 35), AsA-$[C]_n$ is constantly being consumed, while the total content of (+)-catechin extension unit involved in the polymerization is increasing (Fig. 9). Combined with the evidence for non-enzymatic polymerization of AsA-$[C]_n$ with (−)-epicatechin in vitro, the results from grape indirectly support the possibility that (+)-catechin PA extension unit is the destination of AsA-$[C]_n$ when flavan-3-ol starter units are present in vivo. Considering its reactivity, leucocyanidin is still a more direct (+)-catechin carbocation doner than the (+)-catechin conjugates for non-enzymatic PA extension. Thus, the presence of AsA-$[C]_n$ more like represents a buffer pool of excess leucocyanidin for pre-assembly of (+)-catechin extension unit before participating in PA polymerization.

## Discussion

Flavan-3-ol monomers (primarily (−)-epicatechin or (+)-catechin), and flavan-3-ol intermediates with reactive C4 position,

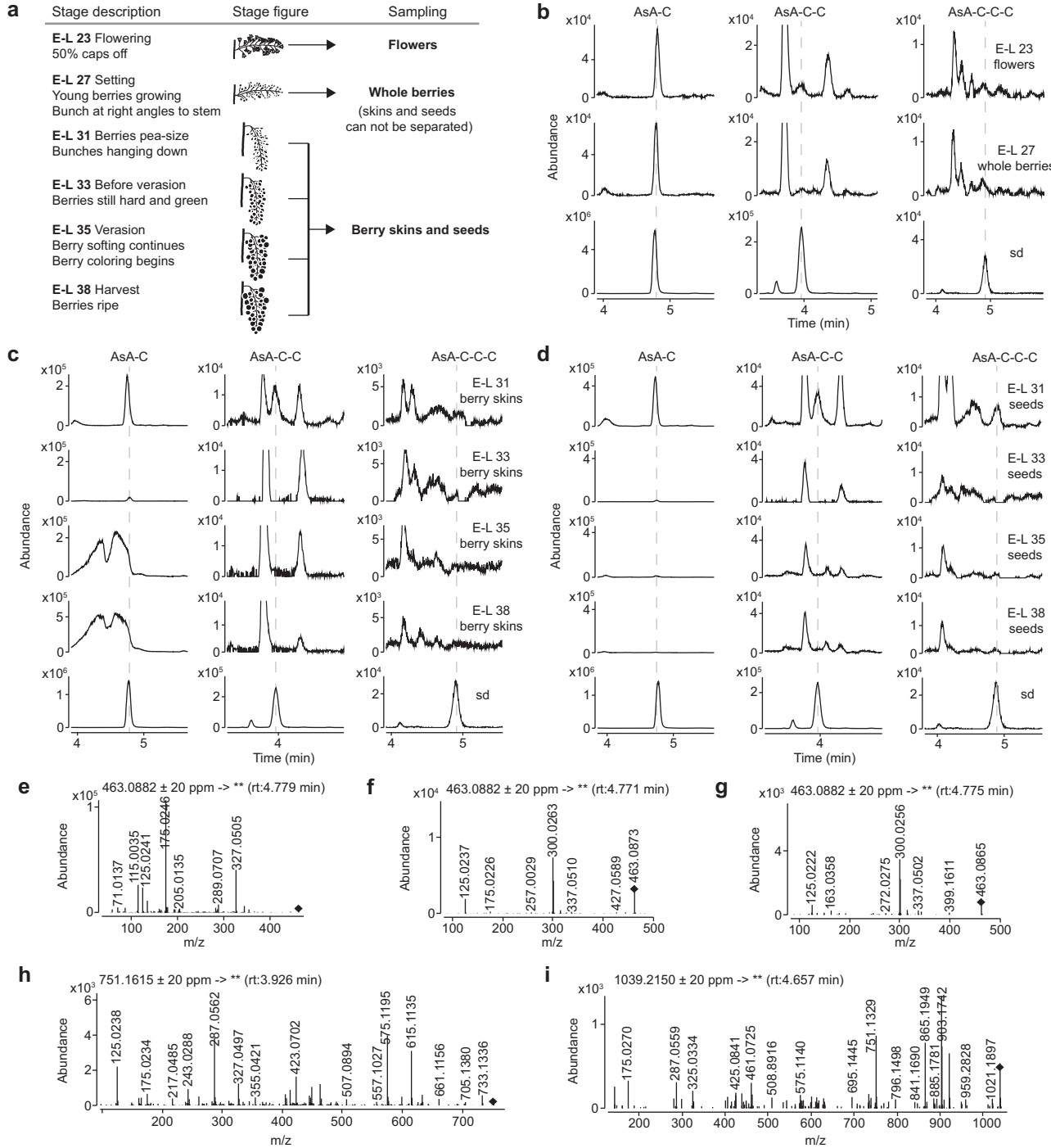

**Fig. 7 AsA-[C]$_n$ exists in flowers, berry skins, and seeds of grapevine at the stages of active PA biosynthesis. a** Stages analyzed. **b** Targeted detection of AsA-C (EIC at $m/z$ 463.0882 ± 20 ppm), AsA-C-C (EIC at $m/z$ 751.1615 ± 20 ppm) and AsA-C-C-C (EIC at $m/z$ 1039.2150 ± 20 ppm) in grape flowers (E-L 23) and setting berries (E-L 27) using UHPLC-QToF. **c** Targeted detection of AsA-[C]$_n$ in grape berry skins at stages E-L 31, E-L 33, E-L 35, and E-L 38 using UHPLC-QToF. **d** Targeted detection of AsA-[C]$_n$ in grape seeds at stages E-L 31, E-L 33, E-L 35, and E-L 38 using UHPLC-QToF. Product ion profiles of the target compounds in grape berry skins at E-L 31 are shown in (**e**), (**h**) and (**I**) respectively. **f**, **g** The product ions of $m/z$ 463.0882 ± 20 ppm in grape berry skins at E-L 35 and E-L 38 indicate that the humped peaks in (**c**) do not contain AsA-C. The product of the reaction of AsA + leucocyanidin (Leu) serves as the standard (sd). rt, retention time.

serve as the starter and extension units, respectively, for PA polymerization via nucleophilic attack[1,8]. Here we show that the *A. thaliana ans* null mutant, which makes high levels of (+)-catechin-type extension units but no (−)-epicatechin-based PAs, contains AsA conjugates of monomeric and oligomeric (+)-catechin (AsA-[C]$_n$) that do not exist in the wild-type. This

group of compounds also accumulates in berry tissues of grapevine during active PA biosynthesis, suggesting that AsA-[C]$_n$ formation is not just a special case in an *ans* mutant.

For decades, (+)-catechin-type PA units were proposed to be sourced from leucocyanidin, a highly reactive intermediate synthesized by DFR[9]. The downstream enzyme ANS can metabolize

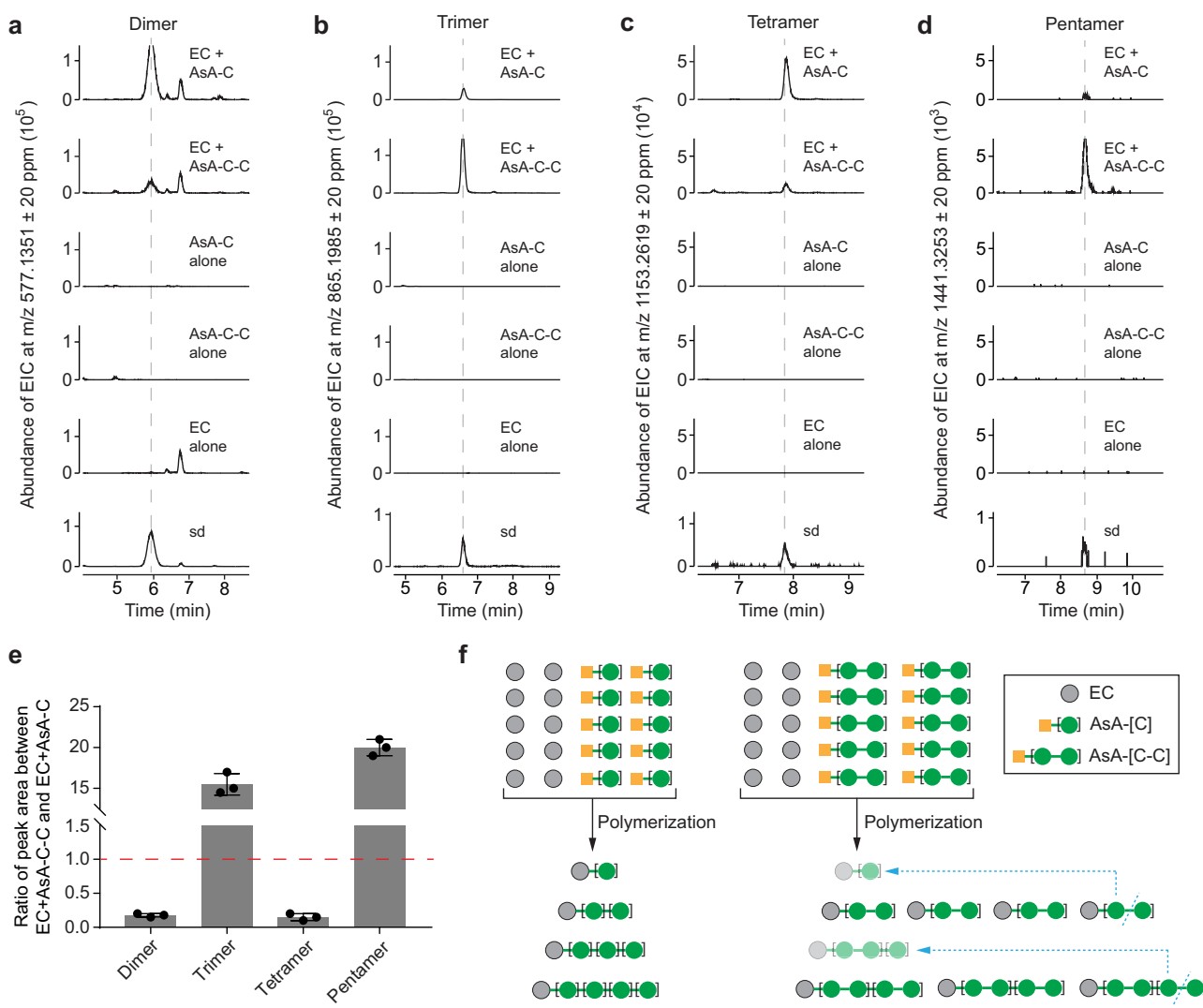

**Fig. 8 In vitro non-enzymatic polymerization of (−)-epicatechin (EC) with either AsA-C or AsA-C-C. a** Detection of PA dimer formation (EIC at $m/z$ 577.1351 ± 20 ppm). **b** Detection of PA trimer formation (EIC at $m/z$ 865.1985 ± 20 ppm). **c** Detection of PA tetramer formation (EIC at $m/z$ 1153.2619 ± 20 ppm). **d** Detection of PA pentamer formation (EIC at $m/z$ 1441.3253 ± 20 ppm). **e** The ratio of the peak areas of the different PAs between the EC + AsA-C-C and EC + AsA-C reaction groups. Data are shown as the mean ± SD (for $n = 3$ independent replicates, paired comparisons between two reaction groups). **f** A model of AsA-C and AsA-C-C participating in PA polymerization as extension units. During PA oligomerization, AsA-C acts as the monomeric extension unit by sequentially adding a single building block, while AsA-C-C acts as the dimeric extension unit by sequentially adding two building blocks at once. Blue dashed lines with arrows indicate that PA by-products with even number degree of polymerization (DP) in EC + AsA-C-C are possibly sourced from the depolymerization of PAs with odd number DP. Source data of Fig. 8e are provided as a Source Data file.

leucocyanidin for the further production of (−)-epicatechin-type PA building blocks[13,15]. Grapevine PAs contain both (+)-cate-chin and (−)-epicatechin subunits, whereas *A. thaliana* exclu-sively uses (−)-epicatechin as the PA building block. In contrast with grapevine, it is believed that leucocyanidin does not "leak out" of a coupled DFR/ANS reaction in *A. thaliana*[11]. Compared with the *A. thaliana ans* null mutant, a weak *ans* mutant pos-sessed much lower amounts of AsA-[C]$_n$, indicating that the biosynthesis of AsA-[C]$_n$ is closely related to the production of leucocyanidin. In addition, down-regulating AsA biosynthesis in an *ans-4* mutant resulted in decreased leucocyanidin production and lower AsA-[C]$_n$ level in developing siliques. AsA is tightly linked to flavonoid metabolism. F3H and ANS (the enzymes upstream and downstream of DFR respectively) require high concentrations of AsA for full catalytic activity, suggesting that the F3H-DFR-ANS metabolon is surrounded by high subcellular concentrations of AsA[32,38,39]. As these enzymes are cytosolically localized[17,40,41], there is a high chance that leucocyanidin not

consumed by ANS will encounter AsA nearby in the neutral pH environment of the cytosol. AsA-[C]$_n$ can be generated by incubating AsA with leucocyanidin at pH 7.4, suggesting that the biosynthesis of these ascorbate-linked polymers is non-enzymatic. Cys, another nucleophile participating in the PA pathway, trap-ped much less leucocyanidin than AsA and the conjugates could not further form PA-like oligomers. The acidic dissociation constant (p$K_a$) of AsA (3-O-H) in water is about 4.1, while that of Cys (S-H) is around 8.5[42,43]. Thus, compared with Cys, AsA more easily donates a hydrogen ion to become nucleophilic at the pH of the cytosol. AsA is therefore a suitable nucleophilic starter unit for leucocyanidin-derived carbocation polymerization in vivo.

Wang et al.[8] trapped PA carbocations from plant materials in the form of (epi)catechin conjugates of nucleophile extraction buffers, leading us to double-check whether AsA-[C]$_n$ exists in planta before extraction. WT *A. thaliana* possesses only (−)-epicatechin carbocation and its *ans* mutant contains only

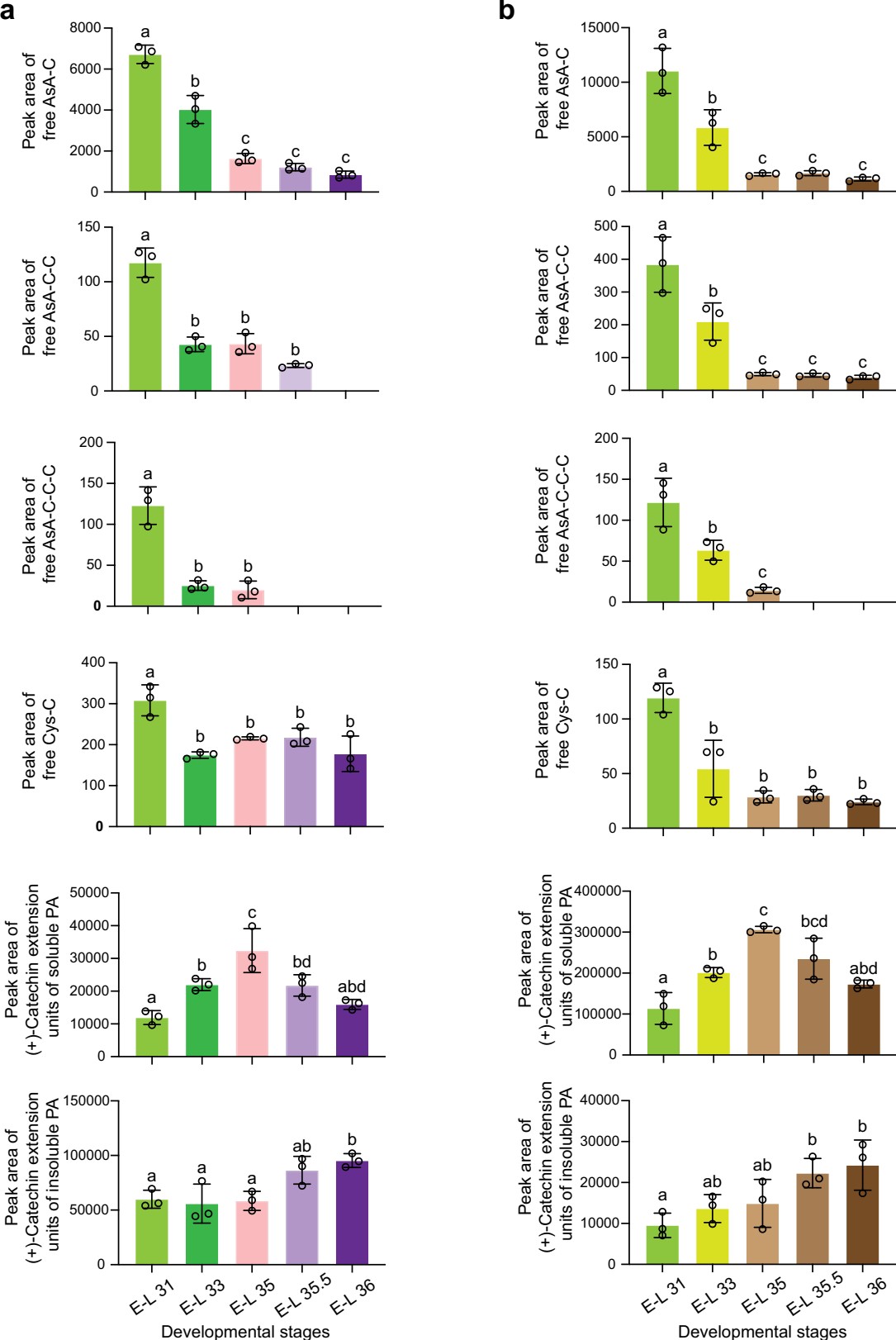

(+)-catechin carbocation[8], and the carbocation level in WT seeds is much higher than that of the *ans* mutant. If AsA traps flavan-3-ol carbocations (either from free carbocation or released from PA cleavage) during the extraction process, the formation of AsA-[EC]n will be more efficient in extracts from WT than that of AsA-[C]n after extraction of the *ans* mutant. However, our LC/MS evidence showed that neither AsA-[C]n nor AsA-[EC]n was present in WT extracts, indicating that the extraction process did not artifactually introduce the AsA conjugates of flavan-3-ol, and that AsA-[C]n is naturally formed in planta. Using thiolysis in the presence of excess Cys, the average length (or mDP) of soluble AsA-[C]n in the *A. thaliana ans-4* mutant was calculated as 5,

**Fig. 9 Levels of AsA-[C]ₙ, Cys-C, and (+)-catechin-type extension unit involved in PA polymerization in grape berry skins and seeds at different E-L developmental stages before harvest. a** Levels of AsA-[C]$_n$, Cys-C and (+)-catechin-type extension unit involved in PA polymerization in grape berry skins. **b** Levels of AsA-[C]$_n$, Cys-C and (+)-catechin-type extension unit involved in PA polymerization in grape seeds. MRM transitions of *m/z* (463.1 → 175.0), *m/z* (751.0 → 287.0), *m/z* (1039.0 → 863.0) and *m/z* (408.1 → 125.0) were used for the detection of AsA-C, AsA-C-C, AsA-C-C-C and Cys-C respectively on UHPLC-QqQ. (+)-Catechin-type extension unit in the soluble PAs after Sephadex LH-20 resin purification and insoluble PAs were represented by Cys-C after thiolysis with the excess of Cys. Levels of the compounds are represented by the peak area on UHPLC-QqQ. Data are shown as the mean ± SD (for *n* = 3 biologically independent samples; the different letters above the bars represent statistically significant differences (at *P* < 0.05) determined by one-way ANOVA with Tukey's multiple comparisons test). Source data of Fig. 9a and b are provided as a Source Data file.

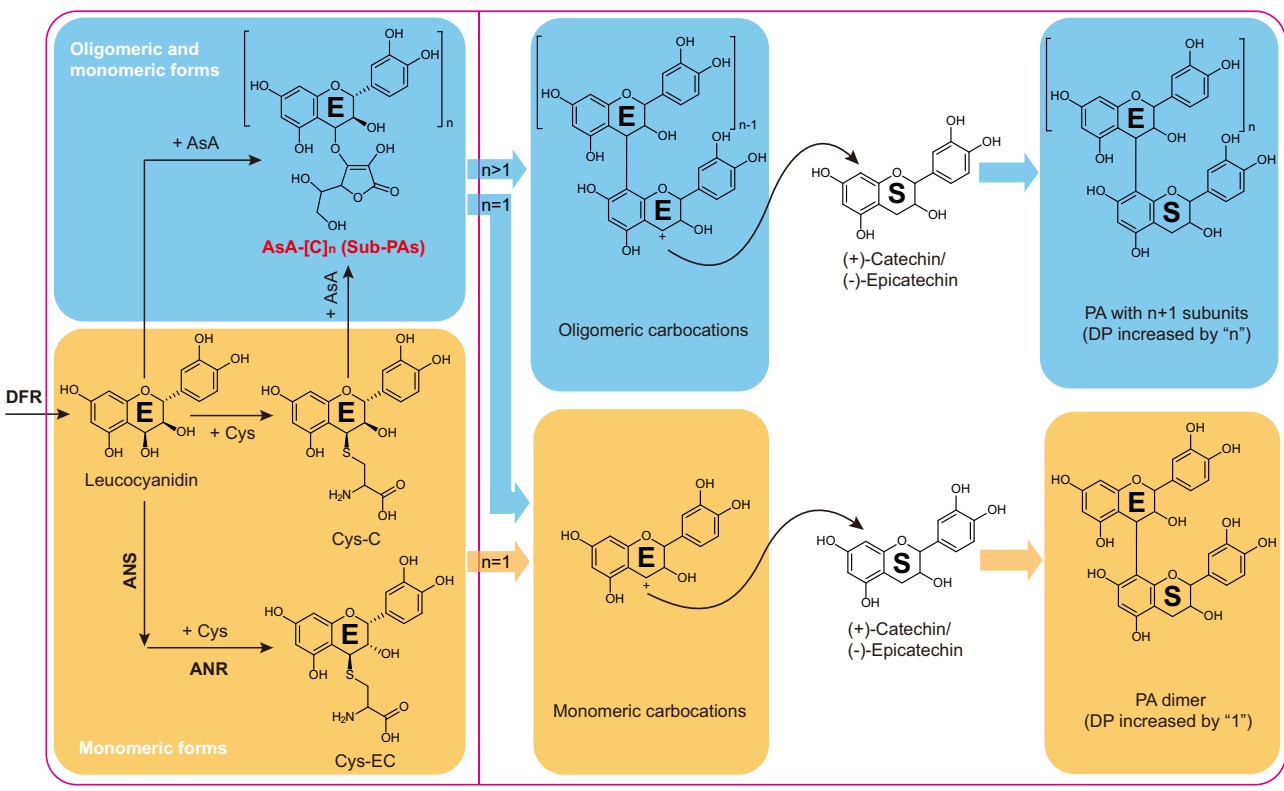

**Fig. 10 An updated model for PA polymerization.** PA extension units are denoted as "E" and PA starter units are denoted as "S". The arrows without enzymes indicate the non-enzymatic pathway. Cys, cysteine; AsA, ascorbate. DP, degree of polymerization.

consistent with the AsA-C (dimer) to AsA conjugate with seven (+)-catechin subunits (octamer) trapped by mass spectrometry of the soluble fraction. AsA-C and (+)-catechin-type extension units were released from the insoluble fraction of the *ans-4* mutant after thiolysis, further suggesting that AsA-[C]$_n$ with poor solubility exists. The concentration ratio between the total extension unit calculated from butanolysis and total AsA-C (including free soluble AsA-C and that released from polymers after thiolysis) was as high as 113 in the *ans-4* mutant, indicating that unconventional PA-like molecules other than AsA-[C]$_n$ may potentially exist. However, we could not rule out a role of AsA to initiate polymerization of very long chain and eventually insoluble polymers, which are believed to store a good portion of the total extension units.

By incubating with (−)-epicatechin monomer at neutral pH, we have shown that AsA-C can donate a monomeric PA extension unit, whereas AsA-C-C can provide two extension units at once during PA polymerization. The mechanism might be similar to that proposed for Cys-C and Cys-EC[6,11,15] (Fig. 10), as both AsA and Cys moieties conjugate with C4 of flavan-3-ols via chalcogens. In the *A. thaliana ans* mutant, AsA-[C]$_n$ might only

function as an alternative PA extender when flavan-3-ol starter units are absent. In contrast, grape possesses (+)-catechin and (−)-epi(gallo)catechin monomers as the classical PA starter units[18]. During berry development, PA biosynthesis starts at flowering and ends by the veraison stage when both the content and DP of PAs reach their peaks[34,44]. Our LC/MS analysis showed that AsA-[C]$_n$ only accumulated during active PA biosynthesis in grape flowers, berry skins and seeds. In both skins and seeds, the AsA-[C]$_n$ level continuously decreased as the amount of (+)-catechin extension of PAs increased. Unlike in the *A. thaliana ans* mutant, AsA-[C]$_n$ was not detected in the insoluble fraction of grape berry tissues, further indicating that grape AsA-[C]$_n$ may participate in PA polymerization before the molecular weight becomes sufficiently large to render the polymers insoluble. In both berry skins and seeds, the continuous decrease in Cys-C level stopped earlier than the decrease in AsA-C, when the content of (+)-catechin extension units in PAs was still increasing, suggesting that Cys-C plays a more transient role than AsA-C in supplying (+)-catechin extension units for PA polymerization. The formation of Cys-C is less favorable than that of AsA-[C]$_n$ in vivo, consistent with the fact that the

abundance of Cys-C in grape berry skins and seeds was below the quantification limit of the current LC/MS equipment. Furthermore, based on in vitro chemistry, the Cys moiety in Cys-C tends to be replaced by AsA, which may also be plausible in vivo because of the high subcellular concentration of AsA and its higher nucleophilicity. In this scenario, Cys-C might act as a transporter of the leucocyanidin backbone to add (+)-catechin extension units to AsA, as shown in the model presented in Fig. 10. The BDE predictions showed that the C4-O bond of AsA-$[C]_n$ ($n > 1$) was slightly weaker than that of AsA-C and much weaker than the C-S bond in Cys-C, suggesting that AsA-$[C]_n$ can more easily donate carbocations than Cys-C for PA polymerization, reflecting the relatively low level of AsA-$[C]_n$ ($n > 1$) in grape. Based on the above evidence, we infer that AsA-$[C]_n$ might play a more important role than Cys-C as a pool of PA extension units for non-enzymatic polymerization.

Combining the findings in *A. thaliana* and grapevine, we propose a two-step (+)-catechin-type PA polymerization mechanism in which temporal or cellular environmental factors prevent all leucocyanidin from integrating into PAs in wild-type plants: AsA first traps leucocyanidin or other (+)-catechin-type reactive intermediates to form pre-assembled oligomeric intermediates in a relatively stable form, and these AsA conjugates themselves can further provide extension units for classical PA polymerization when they encounter flavan-3-ol monomers (Fig. 10). The first polymerization step, especially in the *A. thaliana ans* mutant, reflects the ability of AsA conjugates of (+)-catechin to stabilize the reactive leucocyanidin, which is believed to be toxic to plants[45]. The latter polymerization step is consistent with the hypothesis of a pool for the storage of PA extension units[11]. In view of the fact that AsA-$[C]_n$ not only exists as an oligomer, but also serves as a buffer pool of PA extension units, we suggest that this group of compounds be called sub-proanthocyanidins (sub-PAs).

## Methods

**Chemicals**. (+)-Catechin and (−)-epicatechin were purchased from YuanYe Biotechnology (Shanghai, China). Procyanidin B2, procyanidin B3 and (+)-dihydroquercetin were purchased from Adooq BioScience (Irvine, CA, USA). Cysteine and L-ascorbate were purchased from Sigma Aldrich (St Louis, MO, USA).

**Plant materials**. The seeds of *Arabidopsis thaliana* mutant ans-4 (SALK_073183) and ans-2 (SALK_028793) were gifts from Professor Ming Zhao at the Institute of Crop Science, Chinese Academy of Agricultural Sciences. The seeds of *A. thaliana* mutant vtc2-5 (SALK_146824) were obtained from the Arabidopsis Biological Resource Center at Ohio State University (http://abrc.osu.edu). The seeds of *A. thaliana* wild-type (Columbia-0) were kindly provided by Professor Dapeng Zhang at Tsinghua University. vtc2-5:ans-4 double mutant was constructed by genetic crossing. The homozygous T-DNA insertion lines were verified by PCR using the primers listed in Supplementary Table 2. Dry mature seeds were first placed on wet filter paper in the dark at 4 °C for 3 d and then transferred to soil, with a 16 h light/ 23 °C and 8 h dark/21 °C photoperiod in the greenhouse. Developing siliques at 7 DAP were harvested and frozen immediately in liquid nitrogen. Grape flowers and berries at the indicated stages were sampled in 2018 on grapevine 'Vitis vinifera L. cv. Cabernet Sauvignon' (roots planted in 2010) in a commercial vineyard in Rushan, Shandong, China (36°57' North, 121°30' East, 30 m Altitude). Different berry-derived tissues were frozen in liquid nitrogen immediately after separation from the fresh samples. Frozen samples of Arabidopsis and grape were stored at −80 °C until analysis.

**Determination of PA content**. One hundred mg of frozen samples were ground into a fine powder under liquid nitrogen. The powder was extracted with 1 mL of 70% (v/v) acetone/water (with 0.5% (v/v) acetic acid) by sonication in an ice water bath for 30 min. The resulting mixture was centrifuged at 12,000 × g for 5 min and the supernatant was collected. The sediments were re-extracted twice and all the supernatants were pooled in a 15 mL tube. The pellets after acetone/water extraction contained insoluble PAs and were stored at −20 °C until use. Three mL chloroform was added to the pooled supernatants followed by vigorous vortexing. After centrifuging at 4000 × g for 10 min, the upper phases were collected and further extracted twice with chloroform. The resulting aqueous solutions (soluble PA fraction) were lyophilized and redissolved in 50% (v/v) methanol/water for LC/

MS analysis or soluble PA quantification. Soluble and insoluble PAs were quantified by the dimethylaminocinnamaldehyde (DMACA) method and the butanolysis method respectively[11], with slight modifications. Briefly, soluble PA fraction (5 μL) was mixed with 100 μL of 0.2% (w/v) DMACA in methanol/HCl (1:1, v/v), and the absorbance at 640 nm was measured after 4 min at room temperature. DMACA reaction with (+)-catechin was processed in parallel and served as standard. The insoluble fraction was resuspended with 1 mL of butanol/HCl (95:5, v/v). To quantify insoluble PAs, the slurries were sonicated in ice water for 30 min and centrifuged at 12,000 × g for 5 min. One hundred μL aliquots of the supernatants were applied to the wells of a 96-well plate for recording the absorption at $A_{550}$, and then transferred back to the reaction tubes followed by heating at 95 °C for 1 h. After cooling to room temperature, the mixtures were vigorously vortexed and centrifuged at 12,000 × g for 5 min. One hundred μL supernatants were measured again at $A_{550}$ and the values from the previous readings subtracted. Butanolysis of procyanidin B2 was processed in parallel and served as standard.

To prepare the samples for soluble PA extension unit measurement, soluble PAs extracted from 100 mg ground samples were all subjected to lyophilized. And the insoluble fraction after soluble PA extraction were directly used for further analysis of insoluble PA extension units. To prepare the samples for total PA extension unit quantification, 100 mg ground powder was extracted with 1 mL chloroform by sonicating in ice water for 30 min and the supernatants were discarded after centrifugation at 12,000 × g for 5 min. The pellets were re-extracted with chloroform twice and stored at −20 °C until analysis. To quantify and visualize PA extension units, the lyophilized soluble PAs, the sediments after soluble PA extraction, and the samples with chlorophyll removed by chloroform but without acetone/water extraction were resuspended with 1 mL of butanol/HCl (95:5, v/v). The butanolysis and quantification methods were exactly the same as that of insoluble PA measurement described above. For visualization, 500 μL of supernatant after butanolysis was transferred to a 1.5 mL microtube for photographic capture of the red pigmentation. Butanolysis of procyanidin B3 was processed in parallel and served as standard.

**Purification of grape PAs with Sephadex LH-20 resin**. Grape PAs for analyzing (+)-catechin extension unit level were purified by the published Sephadex LH-20 resin method[16], with slight modifications. Briefly, to extract soluble PAs, frozen grape berry skins and seeds samples were ground into a fine powder under liquid nitrogen followed by lyophilization. Twenty mg of dry powder was extracted with 1 mL of 70% (v/v) acetone/water (with 0.5% (v/v) acetic acid) for three times, and the pooled supernatants were then extracted with equal volumes of chloroform for three times as described above. One hundred μL of soluble PAs were lyophilized and re-dissolved in 100 μL of 50% (v/v) methanol/water followed by the DMACA quantification. Thirty μL of soluble PAs in 50% (v/v) methanol/water (containing no more than 20 μg soluble PAs) were mixed with 150 μL of a slurry of Sephadex LH-20 resin equilibrated with 50% (v/v) methanol/water (dry powder, 4 mL g⁻¹) in a 2 mL microtube for 10 min incubation with rotation. The resin was then washed twice with 1 mL of 50% (v/v) methanol/water and eluted with 200 μL of 50% (v/v) acetone/water for three times. The eluted PA fractions were dried under nitrogen flow for further thiolysis with the excess of Cys.

**Thiolysis and calculation of mean degree of polymerization (mDP)**. Fifty μL of soluble PA aliquot was lyophilized and re-dissolved in 100 μL of thiolysis buffer with excess Cys (18 mg mL⁻¹ Cys dissolved in 0.5 N HCl in methanol)[6] followed by incubation at 50 °C for 30 min. To terminate the reaction and remove the HCl in the lysis buffer that is harmful for mass spectrometry, the reaction system was dried by nitrogen flow and then re-dissolved in 100 μL 50% (v/v) methanol/water. To prepare the sample for quantification of products before lysis, 50 μL of soluble PA aliquot was lyophilized and re-dissolved directly in 100 μL 50% (v/v) methanol/ water. The products before and after thiolysis were subjected to UHPLC-QqQ for quantification. The mDP of PAs in the WT and AsA-$[C]_n$ in the ans-4 mutant were calculated according to Eqs. (1) and (2) respectively.

$$\text{mDP} = \frac{(\text{Flavan-3-ol}_{\text{after lysis}} - \text{Flavan-3-ol}_{\text{before lysis}}) + (\text{Cys-EC}_{\text{after lysis}} - \text{Cys-EC}_{\text{before lysis}})}{\text{Flavan-3-ol}_{\text{after lysis}} - \text{Flavan-3-ol}_{\text{before lysis}}}$$

(1)

$$\text{mDP} = \frac{\text{AsA-C}_{\text{after lysis}} \times 2 + (\text{Cys-C}_{\text{after lysis}} - \text{Cys-C}_{\text{before lysis}})}{\text{AsA-C}_{\text{after lysis}}}$$

(2)

To analyze the PA composition in the insoluble fraction of 7 DAP siliques, the pellets after soluble PA extraction from 100 mg fresh weight material were resuspended in 500 μL thiolysis buffer with excess Cys[6] and incubated at 50 °C for 30 min. The reaction mixture was then centrifuged at 12,000 × g for 5 min and the supernatant was collected. The sediments were re-lysed twice and all the supernatants were pooled. Fifty μL of the pooled supernatant was dried by nitrogen flow and re-dissolved in 100 μL 50% (v/v) methanol/water. The thiolysis products were analyzed by UHPLC-QqQ.

**Preparation of crude polyphenol extracts for leucocyanidin measurement**. The crude polyphenol extracts from *A. thaliana* siliques were prepared using our previously published method[16], with slight modifications. Briefly, fresh samples

were ground in liquid nitrogen, and 500 μL of 80% (v/v) methanol/water was added to 100 mg of ground samples followed by 15 min of sonication at 4 °C in the dark. The mixture was centrifuged for 5 min at 16,000 × g and the supernatant was immediately subjected to measurement of leucocyanidin with UHPLC-QToF as described below.

**LC/MS analysis**. Targeted analysis of (+)-catechin/ (−)-epicatechin and B-type procyanidin oligomers in *A. thaliana* was conducted using an Agilent 1200 HPLC system equipped with a 6410 triple quadrupole mass spectrometer (HPLC-QqQ) (Agilent). An Agilent Poroshell 120 EC column (2.1 × 150 mm, 2.7 μm, C18) was used for metabolites separation. The method for multiple reaction monitoring (MRM) in negative mode on HPLC-QqQ was developed previously[46]. In detail, the HPLC elution program was as follows: solvent A (0.1% [v/v] formic acid in water) and solvent B (0.1% [v/v] formic acid in acetonitrile/ methanol (1:1, v/v)); flow rate, 0.4 mL min⁻¹; gradient, 0–28 min, 10–46% B; 28–29 min, 46–10% B. For detection of mass spectra in negative mode, ion source parameters were as follows, gas/ source heater, 350 °C/150 °C; gas flow, 12 L min⁻¹; nebulizer, 35 psi; capillary, 4 kV. MRM parameters were as follows: for (+)-catechin and (−)-epicatechin, the transition of $m/z$ (289 → 123) was monitored with fragmentor (Frag) 115 V and collision energy (CE) 24 eV; for procyanidin dimer, the transition of $m/z$ (577 → 407) was monitored with Frag 165 V and CE 22 eV; for procyanidin trimer, the transition of $m/z$ (865 → 407) was monitored with Frag 145 V and CE 22 eV. Dwell 135 ms was set for all the compounds. The neutral loss scan of 288 was also performed on the above HPLC-QqQ instrument using the precursor ion scan range $m/z$ 300–1000, Frag 135 V, and CE 20 eV, with the same HPLC elution program and ion source parameters described above. The product ion profiles for the candidate compounds on HPLC-QqQ were obtained by using Selected Ion Monitoring followed by product ion scan (SIM-Scan) with fragmentor 135 V and collision energy 20 eV.

UHPLC-QToF analysis was carried out on an Agilent 1290 Infinity II UHPLC system coupled with a 6546 QToF (Agilent). An Agilent Zorbax RRHD SB column (3 × 150 mm, 1.7 μm, C18) was used for metabolite separation. The UHPLC elution program was as follows: solvent A (0.1% [v/v] formic acid in water) and solvent B (0.1% [v/v] formic acid in acetonitrile); flow rate, 0.4 mL min⁻¹; gradient, 0–1 min, 5% B; 1–2 min, 5–10% B; 2–17 min, 10–31% B; 17–18 min, 31–95% B; 19–21 min, 5% B. For detection of mass spectra in negative mode, ion source parameters were as follows, gas/sheath gas heater, 350 °C/400 °C; gas flow/sheath gas flow, 8 L min⁻¹/10 L min⁻¹; nebulizer, 40 psi; capillary, 4 kV. The scan range for MS1 was $m/z$ 100–1700 with sampling frequency at 4 spectra/s and the scan range for MS2 was $m/z$ 100 to (mass of candidate + 50) with sampling frequency at 4 spectra/s. The fragmentor was set as 150 V and collision energy ranged from 20 to 30 eV.

Based on the above UHPLC-QToF method, some parameters were optimized for the detection of the unstable leucocyanidin. The UHPLC elution program for leucocyanidin detection was as follows: solvent A (0.1% [v/v] formic acid in water) and solvent B (0.1% [v/v] formic acid in acetonitrile); flow rate, 0.4 mL min⁻¹; gradient, 0–1 min, 5% B; 1–2 min, 5% to 10% B; 2–10 min, 10–21% B; 10–12 min, 21–95% B; 12–14 min, 5% B. The scan range for MS1 was $m/z$ 50–500 with sampling frequency at 4 spectra/s and the scan range for MS2 was $m/z$ 50–500 with sampling frequency at 4 spectra/s. The fragmentor and the collision energy were set as 150 V and 10 eV respectively. For the auto-sampler module of the Agilent 1290 Infinity II UHPLC system, the vial plate temperature was set as 10 °C and the illumination was turned off. The area of extracted ion chromatogram (EIC) at 305.0667 ± 20 ppm was used for leucocyanidin quantification from the standard curve of developed by the chemical synthesized standard.

The quantification of products before and after thiolysis was conducted using an Agilent 1290 Infinity II UHPLC system coupled with a 6470B triple quadrupole mass spectrometer (UHPLC-QqQ) (Agilent). The UHPLC system was equipped with an Agilent Zorbax Eclipse Plus column (2.1 × 50 mm, 1.8 μm, C18) or an Agilent Poroshell 120 SB column (2.1 × 150 mm, 2.7 μm, C18) for metabolite separation. The UHPLC elution program for the Agilent Zorbax Eclipse Plus column was as follows: solvent A (0.1% [v/v] formic acid in water) and solvent B (0.1% [v/v] formic acid in acetonitrile); flow rate, 0.2 mL min⁻¹; gradient, 0–0.5 min, 5% B; 0.5–1.5 min, 5–10% B; 1.5–6.5 min, 10–20% B; 6.5–7.5 min, 20–95% B. And the UHPLC elution program for the Agilent Poroshell 120 SB column was as follows: solvent A (0.1% [v/v] formic acid in water) and solvent B (0.1% [v/v] formic acid in acetonitrile); flow rate, 0.4 mL min⁻¹; gradient, 0–1 min, 5% B; 1–2 min, 5–10% B; 2–9 min, 10–20% B; 9 to 10 min, 20–95% B. The MS parameters in the negative mode were optimized following the manufacturer's handbook using (+)-catechin, Cys-C, AsA-C, AsA-C-C, and AsA-C-C-C as standards. Ion source parameters were as follows: gas/sheath gas heater, 300 °C /325 °C; gas flow/sheath gas flow, 7 L min⁻¹/11 L min⁻¹; nebulizer, 30 psi; capillary, 3.5 kV; nozzle voltage 1500 V. MRM parameters were as follows: for Cys-C and Cys-EC, the transition of $m/z$ (408.1 → 125.0) was monitored with fragmentor (Frag) 95 V and collision energy (CE) 20 eV; for (+)-catechin and (−)-epicatechin, the transition of $m/z$ (289.1 → 123.1) was monitored with Frag 130 V and CE 35 eV; for AsA-C, the transition of $m/z$ (463.1 → 175.0) was monitored with Frag 100 V and CE 15 eV; for AsA-C-C, the transition of $m/z$ (751.0 → 287.0) was monitored with Frag 135 V and CE 30 eV; for AsA-C-C-C, the transition of $m/z$ (1039.0 → 863.0) was monitored with Frag 135 V and CE 20 eV; dwell 150 ms and cell acceleration 5 V were set for all the compounds.

The detection of AsA-[C]ₙ pentamer to octamer was performed by the same UHPLC-QqQ with the same ion source parameter. An Agilent Zorbax Eclipse Plus column (2.1 × 50 mm, 1.8 μm, C18) was used to separate metabolites. The UHPLC elution program was as follows: solvent A (0.1% [v/v] formic acid in water) and solvent B (0.1% [v/v] formic acid in acetonitrile); flow rate, 0.2 mL min⁻¹; gradient, 0–0.5 min, 5% B; 0.5–1.5 min, 5–10% B; 1.5–11.5 min, 10–30% B; 11.5–16.5 min, 30–50% B; 16.5–17.5 min, 50–90% B. MRM parameters were as follows: for pentamer, the transition of $m/z$ (1327.3 → 1151.2) was monitored with fragmentor (Frag) 150 V and collision energy (CE) 25 eV; for hexamer, the transition of $m/z$ (1615.3 → 1151.2) was monitored with Frag 135 V and CE 35 eV; for heptamer, the transition of $m/z$ (1904.3 → 1151.2) was monitored with Frag 150 V and CE 35 eV; for octamer, the transition of $m/z$ (2192.5 → 1727.4) was monitored with Frag 150 V and CE 35 eV; dwell 150 ms and cell acceleration 5 V were set for all the compounds.

**Chemical synthesis and purification of flavan-3-ol-conjugates**. Cys-C and Cys-EC were prepared from procyanidin B3 and B2 respectively with Cys under hot acidic hydrolysis conditions followed by HPLC purification as described by Liu et al.[6] with slight modifications. In detail, 200 μg procyanidin standard was dissolved in 50 μL Cys lysis buffer (18 mg mL⁻¹ Cys dissolved in 0.5 N HCl in methanol) and the mixture was incubated at 50°C for 30 min. The reaction was terminated by the addition of 200 μL cold water. An Agilent 1200 HPLC system coupled with a Thermo Fisher HypersilGold column (250 mm × 4.6 mm, 5 μm, C18) was used to purify the target compounds. The HPLC elution program was as follows: solvent A (0.1% [v/v] formic acid in water) and B (0.1% [v/v] formic acid in acetonitrile) at 1 mL min⁻¹ flow rate, 0–5 min, 5% B; 5–10 min, 5–10% B; 10–25 min, 10–17% B; 25–30 min, 17–100% B. The volume per injection was 40 μL, and eluting compounds were detected with a variable wavelength detector (VWD) at 280 nm. Cys-C and Cys-EC were collected at 15 min and 17 min respectively and concentrated by lyophilization. After re-dissolving in 200 μL water, Cys-C and Cys-EC were quantified by HPLC-VWD using (+)-catechin as standard.

2,3-*trans*-3,4-*cis*-Leucocyanidin was synthesized from (+)-dihydroquercetin with a reducing agent followed by acidification and purification as described by Kristiansen[47] with slight modifications. In detail, 2.5 mg (+)-dihydroquercetin and 2.5 mg NaBH₄ were dissolved in 250 μL ethanol. The solution was shaken vigorously at 20°C for 2 h in the dark and then mixed with 5 mL water followed by the pH adjustment to 3.8 with acetic acid. After incubated at 40 °C for 4 h in the dark, the mixture was lyophilized and re-dissolved using 200 μL water. An Agilent 1200 HPLC system equipped with a Waters μBondapak column (30 cm × 3.9 mm, 10 μm, phenyl) was used for compound purification. The elution program for HPLC was as follows: A (water)/B (acetonitrile); 0–24 min, 0% B, 1 mL min⁻¹; 24–27 min, 0–100% B, 2 mL min⁻¹; 27–28 min, 100–0% B, 2 mL min⁻¹; 28–30 min, 0% B, 1 mL min⁻¹. The volume per injection was 35 μL, and products were detected with a variable wavelength detector (VWD) at 280 nm. 2,3-*trans*-3,4-*cis*-Leucocyanidin was collected at 16 min and concentrated by lyophilization. After re-dissolving in 200 μL water, 2,3-*trans*-3,4-*cis*-leucocyanidin was immediately quantified by HPLC-VWD using (+)-catechin as standard and stored at −80 °C until use.

To synthesize AsA-C and AsA-C-C, a mixture of 500 μM 2,3-*trans*-3,4-*cis*-leucocyanidin and 500 μM L-ascorbate (pH 7.4 adjusted by NaOH) was incubated at 30 °C in the dark for 1 h. The reaction was extracted with three volumes of ethyl acetate, which was sufficient to remove the remaining 2,3-*trans*-3,4-*cis*-leucocyanidin as assessed by mass spectrometry. For purification, an Agilent 1200 HPLC system with a variable wavelength detector (VWD) was used. The elution program for the Agilent Zorbax SB preparative column (9.4 × 250 mm, 5 μm, C18) was scaled up from that of the Agilent Zorbax RRHD SB column (3 × 150 mm, 1.7 μm, C18) mentioned above using Waters PrepCalculator (www.waters.com/prepcalculator) with slight modification as follows: solvent A (0.1% [v/v] formic acid in water) and solvent B (0.1% [v/v] formic acid in acetonitrile); flow rate, 3 mL min⁻¹; gradient, 0–5 min, 5% B; 5–10 min, 5–10% B; 10–20 min, 10–15% B; 20–23 min; 15–95% B. The volume per injection was 100 μL, and eluting compounds were detected at 280 nm. AsA-C and AsA-C-C eluted at 17 and 15 min respectively. The corresponding fractions were collected and concentrated by lyophilization. After re-dissolving in water, AsA-C and AsA-C-C were quantified by HPLC-VWD using (+)-catechin and procyanidin B3 as standards respectively.

**Non-enzymatic PA condensation in vitro**. Non-enzymatic reactions were carried out in 100 μL reaction volumes including 50 mM Tris-HCl buffer (pH 7.4) and the combination of two substrates (100 μM of each) as follows: Cys + 2,3-*trans*-3,4-*cis*-leucocyanidin, AsA + 2,3-*trans*-3,4-*cis*-leucocyanidin, AsA + Cys-EC, AsA + 2,3-*trans*-3,4-*cis*-leucocyanidin, (−)-epicatechin + AsA-C, (−)-epicatechin + AsA-C-C. The mixtures were incubated at 30 °C in the dark for 1 h. (−)-Epicatechin + AsA-C and (−)-epicatechin + AsA-C-C reactions were terminated by addition of three volumes of ethyl acetate. The ethyl acetate phase was dried under nitrogen and the residue re-dissolved in 100 μL water for UHPLC-QToF analysis. The other reactions were subjected to targeted MS/ MS analysis directly after incubation.

**Prediction of bond dissociation energies**. The chemical structures were drawn with InDraw (http://indrawforweb.integle.com/) and the corresponding Simplified Molecular Input Line Entry System (SMILES) strings were exported to ALFABET

BDE Estimator (https://bde.ml.nrel.gov/)[35] for bond dissociation energy predictions.

**Statistical analysis**. Statistical analysis was performed by GraphPad Prism 7 (GraphPad Software Inc.) with two-tailed Student's $t$ test or one-way ANOVA with Tukey's multiple comparisons test. Data were visualized as the average of three biological replicates with SD as error bars on the bar plots.

**Reporting summary**. Further information on research design is available in the Nature Research Reporting Summary linked to this article.

## Data availability

All data needed to evaluate the conclusions in the paper are present in the paper and the Supplementary Information. Raw LC/MS data have been deposited to the EMBL-EBI MetaboLights database (https://www.ebi.ac.uk/metabolights/) with the identifier MTBLS4942. Source data are provided with this paper.

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

## Acknowledgements

The authors thank Dr. Yaqin Wang, Beijing Academy of Agriculture and Forestry Science, for suggestions for mass spectrometry data analysis, and Dr. Xiaotong Gao and Dr. Tianci Shi, China Agricultural University, for assistance with grape sampling and compound extraction respectively. This research was funded by the National Natural Science Foundation of China (Grant U20A2042), China Agriculture Research System of MOF and MARA (CARS-29), and the China Scholarship Council (Grant 201706350125 to K.Y.).

## Author contributions

C.D., R.A.D., and K.Y. conceived and designed the study, and analyzed the data. K.Y. acquired data and wrote the original draft. R.A.D. and C.D. reviewed and edited the final manuscript.

## Competing interests

The authors declare no competing interests.
