## [Peer Review File · Nature Communications]

A role for ascorbate conjugates of (+)-catechin in proanthocyanidin polymerizationREVIEWER COMMENTS

Reviewer #1 (Remarks to the Author):

<Please see attached file>

I have carefully read this manuscript titled “A role for ascorbate conjugates of (+)-catechin in proanthocyanidin polymerization” by Keji Yu et al. The article identified the presence of ascorbate (AsA) conjugates of (+)-catechin in Arabidopsis mutants and grapes through UPLC-QQQ-MS. In addition, the article also mentioned that AsA-C and AsA-C-C-C can participate in the polymerization of PA in vitro. It is an interesting subject in the field.

There have been several articles describing the conjugates (Cys-EC and Cys-C) of (+)-catechin and (-)-Epicatechin carbocations, and they have been proved to be the precursor for the PA condensation reaction (The references “A role for leucoanthocyanidin reductase in the extension of proanthocyanidins, 2016”; “VvLAR1 and VvLAR2 Are Bifunctional Enzymes for Proanthocyanidin Biosynthesis in Grapevine”, 2019). Carbocations are very short-lived and active compounds, which have strong electrophilic properties and are easily attacked by a series of nucleophiles to form conjugates. For example, Peiqiang Wang's literature showed that they can be trapped by methanol, ethanol, isopropanol, and even indole ethanol to form conjugates (The reference “Functional demonstration of plant flavonoid carbocations proposed to be involved in the biosynthesis of proanthocyanidins” 2020). Carbocations are also easily obtained from leucocyanidin and PA. In our opinion, ascorbate conjugate of (+)-catechin is a similar kind of conjugate, coming from the cleavage reaction of leucoanthocyanidin and PA. We speculate that when excessive amount of leucocyanidin are accumulated in plants, in addition to AsA-[C]_n, more other conjugates can be detected in plants. Therefore, the discovery of these compounds in plants is not very surprising.

What we are wondering is, (1) Do these conjugates actually exist in plants or are they produced during the extraction process? After all, leucocyanidin are very unstable, and carbocations are easily produced during the extraction process. (2) If these conjugates actually exist in plants, do they actually participate in the PA condensation reaction? Are they essential intermediates for PA condensation reactions? Though these conjugates can provide carbocations for PA condensation in vitro, it does not indicate that these conjugates are the real substrates of PA condensation reaction in plants. (3) The physiological significance of these conjugates (AsA-C) in plants is worthy of further study, such as whether they have the function of balancing the concentration of carbocations; whether they have the effect on reducing the aggressiveness of carbocations and providing detoxification effects. (4) The author may consider silencing the AsA biosynthesis pathway to prove the significance of these conjugates. Therefore, we feel that the innovation of the paper is still not strong enough to be published in Nature Communications.

In addition, some results in the paper are confusing.

Fig.3 The ion fragment characteristics of AsA-C (m/z- 463) is very similar to your earlier identified epicatechin-glucuronide conjugate (m/z- 463) in 2016, which makes us puzzled.

Line 254. It was described that AsA-[C]_n existed in flowers, berry skins and seeds of grapevine at active stages of PA biosynthesis. So, is there such a possibility that AsA-C was formed during the extraction process, in the process PA cleavage occurred and produced carbocations, which were then trapped by AsA. Combined with your previous papers (“VvLAR1 and VvLAR2 Are Bifunctional Enzymes for

Proanthocyanidin Biosynthesis in Grapevine", 2019), if these conjugates actually exist in grapevine, then in the process of C-type PA polymerization, which of AsA-C and Cys-C played a more important role? It is worth discussing.

Reviewer #2 (Remarks to the Author):

The authors uncover new information on proanthocyanidin (PA) synthesis, specifically identifying the involvement of an ascorbate-catechin conjugate in the polymerisation process and this provides a very significant advance in knowledge. PA is an important plant constituent, so further characterisation of its biosynthesis is of wide interest. Extensive and elegant mass spectrometry analysis and use of an Arabidopsis mutant (ans4) enabled identification of the ascorbate-catechin conjugates, and this further enabled identification of the same conjugate in wild type grape. The authors propose an PA polymerisation pathway involving the ascorbate-catechin conjugate.

The conclusions are supported by extensive data, and I have no specific comments/criticisms on this or the methods, which all appear sound and appropriate. However, given that the journal publishes papers of wider interest, I feel that the authors could improve the abstract, introduction and discussion material to improve clarity for the non-expert. I found the text overall to be quite difficult to follow and a clearer introduction to the pathway would assist in understanding the new data. It would be helpful if the extent to which this conjugate is involved in wild type was stated more clearly in the summary and conclusions.

Although I do not propose that more data are needed for this manuscript, it would be interesting to know to what extent PA polymerisation is influenced by ascorbate concentration (or example in ascorbate deficient (vtc) mutants of Arabidopsis) since there is evidence that its concentration is particularly high in developing siliques (Conklin et al. (2000) Genetics 154(2): 847-856). As an aside (which might not be relevant to reproductive tissue), anthocyanin-related genes have lower expression in leaves at high light intensity in vtc mutants along with less anthocyanin (Page et al. 2012 Plant Cell and Environment 35(2): 388-404) indicating a link between ascorbate synthesis and flavonoid metabolism.

Reviewer #3 (Remarks to the Author):

What are the noteworthy results? These appear to be note worthy in that this area while an important part of plant metabolism, still seems incompletely known and confusing.

Will the work be of significance to the field and related fields? I believe the work is significant as it establishes an import role of ascorbate utilization in plant metabolism

How does it compare to the established literature? If the work is not original, please provide relevant references.. The work is a significant advance

Does the work support the conclusions and claims, or is additional evidence needed? The work is detailed in as much as it attempts to report

Are there any flaws in the data analysis, interpretation and conclusions? Do these prohibit publication or require revision? Not really, just a bit dense in places

Is the methodology sound? Does the work meet the expected standards in your field? yes

Is there enough detail provided in the methods for the work to be reproduced? yes

An excellent paper well written English, describing an important alternative pathway for proanthocyanidin (PA) biosynthesis. However the paper is overly complex skirting some issues such as the source of Cys-EC and Cys-C and ascorbyl [C]n. A clearer introduction to the biochemistry and chemistry of PA synthesis after the known enzymatic steps (LAR, ANS, LDOX and ANR) would be helpful, at least to say that the actual pathway is unclear. The authors should refer to papers on the biosynthesis of PAs earlier in the MS. They do include pertinent literature, but the reviewer had to search for them

However the paper clearly identifies the intermediate ascorbyl catechin as important, although it is not clear whether this intermediate is part of the pathway in wild type plants such as Arabidopsis. Also what is the significance or importance of this ascorbyl-C in wt plants apart from grape?

Line 44. The authors mean synthase, not reductase for ANS

The supplementary figure 1a should be included in the main text rather than as a supplementary figure

Line 100 not sure what only refers to

Fig 1e. statistical analysis should be provided. Explain what the circles mean.

Line 155. Rather than around, a range should be indicated

Equations in line 241 onwards should be in the methods or supplementary

Line 286. The word alone is unclear as to what it refers to

Line 307. There are three chemicals mentioned, but only starter and extension unit referred to in reference to respectively

Line 313. Is there a level of detection?

Line 446. What does cleaned only with chloroform refer to? Unclear

Line 452. Microtube rather than Eppendorf

Line 554. While statistical methods are described the results of this analysis do not seem to be reported

Fig 1 and 2. The WT is above the ana mutant in fig 1 but the reverse is true in fig 2. Be consistent

Fig 6. I would be useful to include the tissue type in the figure to make it clear

Fig 7f. Why are there 5 rows of symbols in this figure?

Response to Reviewer #1

Reviewer #1 (Remarks to the Author):

I have carefully read this manuscript titled “A role for ascorbate conjugates of (+)-catechin in proanthocyanidin polymerization” by Keji Yu et al. The article identified the presence of ascorbate (AsA) conjugates of (+)-catechin in *ans Arabidopsis* mutants and grapes through UPLC-QQQ-MS. In addition, the article also mentioned that AsA-C and AsA-C-C-C can participate in the polymerization of PA *in vitro*. It is an interesting subject in the field.

There have been several articles describing the conjugates (Cys-EC and Cys-C) of (+)-catechin and (-)-Epicatechin carbocations, and they have been proved to be the precursor for the PA condensation reaction (The references “*A role for leucoanthocyanidin reductase in the extension of proanthocyanidins*, 2016”; “*VvLARI and VvLAR2 Are Bifunctional Enzymes for Proanthocyanidin Biosynthesis in Grapevine*”, 2019). Carbocations are very short-lived and active compounds, which have strong electrophilic properties and are easily attacked by a series of nucleophiles to form conjugates. For example, Peiqiang Wang's literature showed that they can be trapped by methanol, ethanol, isopropanol, and even indole ethanol to form conjugates (The reference “*Functional demonstration of plant flavonoid carbocations proposed to be involved in the biosynthesis of proanthocyanidins*” 2020). Carbocations are also easily obtained from leucocyanidin and PA. In our opinion, ascorbate conjugate of (+)-catechin is a similar kind of conjugate, coming from the cleavage reaction of leucoanthocyanidin and PA. We speculate that when excessive amount of leucocyanidin are accumulated in plants, in addition to AsA-[C]_n, more other conjugates can be detected in plants. Therefore, the discovery of these compounds in plants is not very surprising.

Response:

Thank you very much for carefully reviewing our manuscript. The initial reason for us to search unconventional oligomeric PA-like compounds was that the known monomeric PA intermediates or conjugates were not sufficient to help with understanding how plants cope with the available extension units or carbocations when PA starter unit synthesis is blocked, as in the *Arabidopsis ans* mutant or the *Medicago lar:ans* double mutant. And during the compound screening, we inevitably had to make assumptions about possible products based on the knowledge of nucleophilic attack, which is the central mechanism of the final step of PA non-enzymatic polymerization. We agree with you about the idea that other potential conjugates may exist in plants together with AsA-[C]_n and we also mentioned this point in the manuscript.

Different from Cys-C and Cys-EC, the newly found AsA-[C]_n exists as an oligomer, and may also provide oligomeric extension unit for classical PA polymerization. The discovery of AsA-[C]_n (or we term it as “sub-PA”) conveys the notions that (1) PA polymerization is not restricted to addition of single monomeric units to the growing chain; (2) the starter unit to initiate PA oligomerization does not have to be the flavan-3-ol monomer. We look forward the follow-up works inspired by our findings (for example, one step closer to how PA is polymerized at cellular level, or even designing novel PA molecules in plants).

What we are wondering is, (1) Do these conjugates actually exist in plants or are they produced during the extraction process? After all, leucocyanidin are very unstable, and carbocations are easily produced during the extraction process.

Response:

Thank you for raising this concern to help with improving the manuscript.

The idea that (+)-catechin conjugates of ascorbate (AsA) actually exist in plants rather than being produced as artifacts of the extraction is supported by the following evidence. The wild-type (WT) *A. thaliana* possesses PAs exclusively made of (-)-epicatechin (EC) building blocks, whereas the *ans* mutant contains no PAs. In the paper “*Functional demonstration of plant flavonoid carbocations proposed to be involved in the biosynthesis of proanthocyanidins*” (The Plant Journal, 2020), the authors showed that WT *A. thaliana* possesses only (-)-epicatechin carbocation and *ans* mutant contains only (+)-catechin carbocation, and the carbocation level in WT is much higher than that in the *ans* mutant. If AsA traps flavan-3-ol carbocations (either from free carbocation or that from PA cleavage) during the extraction process, the formation of AsA-[EC]_n will be more efficient in WT extracts than that of AsA-[C]_n in extracts of the *ans* mutant. However, our LC/MS evidence showed that neither AsA-[C]_n nor AsA-[EC]_n was present in WT extracts. This means that the extraction process did not introduce artifactual AsA conjugates of flavan-3-ol. Thus, AsA-[C]_n actually exists *in planta*. We have added the relevant discussion in the manuscript. Please see **Lines 404 to 416** in the revised manuscript.

(2) If these conjugates actually exist in plants, do they actually participate in the PA condensation reaction? Are they essential intermediates for PA condensation reactions? Though these conjugates can provide carbocations for PA condensation *in vitro*, it does not indicate that these conjugates are the real substrates of PA condensation reaction in plants.

Response:

Thank you for the questions to give us a direction to further explore the PA polymerization mechanism.

To assess the possibility of AsA-[C]_n involvement in (+)-catechin-type PA polymerization *in vivo*, we further measured AsA-[C]_n levels and the content of (+)-catechin extension units actually involved in PA extension in grape berry skins and seeds at a series of developmental stages. The results showed that, during the PA active stages, AsA-[C]_n level was continuously decreasing, whereas the amounts of (+)-catechin extension unit in PAs kept increasing in both tissues. Such a negative correlation reflects the possible substrate-product relationship between AsA-[C]_n and (+)-catechin PA extension units. In addition, no AsA-C was detected from the lysis products of insoluble PAs of grape tissues, indicating that AsA-[C]_n in grape did not migrate as a whole to insoluble fractions as in the *A. thaliana ans* mutant but rather participate in further reactions. Combined with the evidence for non-enzymatic polymerization of AsA-[C]_n with (-)-epicatechin *in vitro*, the results from grape further indirectly support the possibility that (+)-catechin PA extension unit is the destination of AsA-[C]_n when flavan-3-ol starter units exist *in vivo*. Please see **Lines 337 to 368, Fig. 9, Supplementary Fig. 13 and Supplementary Fig. 14** in the revised manuscript.

To attempt providing direct *in vivo* evidence, we have been repeatedly performing feeding experiments during the past months. As synthesizing enough AsA-[C]_n for soaking plant materials is currently a challenge (time-consuming and really expensive), the strategy we used was to feed (-)-epicatechin to either the developing seeds separated from the siliques of *ans-4* mutant or to young *ans-4* seedlings growing on MS medium supplied with 5% sucrose (AsA-C accumulation can be induced in this system). However, after trying different combination of (-)-epicatechin concentration (from 100 μM to 500 μM) and feeding time (from 20 min to 20 h), we still could not obtain MS/MS information supporting the formation of natural PA oligomers in any experimental system, while non-natural PAs, possibly formed by oxidation of (-)-epicatechin, were detected with the extension of the incubation time. To the best of our knowledge, the *ans* mutant contains all the possible (+)-catechin PA extension units. And no reports suggest that ANS itself participates in enzymatic polymerization of PAs (regardless of whether this mechanism really exists). Thus, it seems implausible that natural PA oligomers were not formed in the feeding experiment. One possibility is that the exogenous (-)-epicatechin may not reach the correct compartment for PA polymerization in the cell. It was shown that significant levels of (-)-epicatechin-glucoside are produced quickly in the plants after feeding (-)-epicatechin. So it is also possible that glucosidation of exogenous (-)-epicatechin might play a more important role than PA polymerization for detoxification in the plant system. We are wondering whether this is the reason why all current evidence in publications supporting non-enzymatic PA polymerization has to rely on *in vitro* studies, including supplying flavan-3-ol monomers to the plant extracts (Wang et al., 2020, *The Plant Journal*, 101(1): 18-36) and incubating flavan-3-ol monomers with

the purified standards as we did in the present study (Liu et al., 2016, *Nature Plants*, 2(12):1-7; Jun et al., 2021, *Science Advances*, 7(20): eabg4682).

Different from the other PA extension units or precursors reported, the discovery of AsA-[C]_n provides a notion that PA oligomerization does not necessarily proceed by sequential addition of a single extension unit. We believe that with advances in cell biology and analytical technology, relevant questions *in vivo* will be largely elucidated in the future by concerted efforts of the whole PA research community.

(3) The physiological significance of these conjugates (AsA-C) in plants is worthy of further study, such as whether they have the function of balancing the concentration of carbocations; whether they have the effect on reducing the aggressiveness of carbocations and providing detoxification effects. (4) The author may consider silencing the AsA biosynthesis pathway to prove the significance of these conjugates. Therefore, we feel that the innovation of the paper is still not strong enough to be published in *Nature Communications*.

Response to (3) and (4):

Thank you. This is an interesting topic to study.

Accordingly, we further crossed the *ans-4* mutant with the AsA deficient mutant *vtc2-5*, which is reported to possess ~70% less total cellular AsA without decreased growth (Becker et al., 2014, *Journal of Experimental Botany*, 65(20): 5903-5918). Double mutant homozygote was obtained, indicating that silencing the AsA biosynthesis pathway is not lethal to the development of seeds with the *ans-4* background. And there was no significant growth difference between *vtc2-5:ans-4* double mutant and *ans-4* single mutant. AsA-[C]_n levels in the siliques of *vtc2-5:ans-4* double mutant were much lower than those in the *ans-4* single mutant, while the Cys-C level was the same between these two mutant lines. The thiolysis experiment showed that the total level of (+)-catechin carbocation in *vtc2-5:ans-4* was significantly lower than that in the *ans-4* single mutant, indicating that AsA deficiency reduced leucocyanidin production. It is known that F3H at the upstream of DFR requires high level of AsA for maintaining the activity (Lukačič and Britsch, 1997, *European Journal of Biochemistry*, 249(3): 748-757). Others have reported that in AsA deficient lines, the expression of anthocyanin biosynthesis related genes (including those encoding CHS and TTG1) is suppressed under high-light conditions (Page et al., 2012, *Plant Cell and Environment*, 35(2): 388-404). Taken together, AsA balanced (+)-catechin carbocation synthesis upstream of AsA-[C]_n, suggesting that detoxification of carbocation is a multi-level process. Considering this, it is nearly impossible to obtain a system in which AsA-[C]_n level is decreased but (+)-catechin carbocation remains the same for providing direct evidence about the physiological significance of AsA-[C]_n. However, we

think the results above address the point as well as we can at present, and provide more details on the AsA-[C]_n biosynthesis network. We have incorporated the new findings in the revised manuscript. Please see **Lines 265 to 284 and Supplementary Fig. 10 to Supplementary Fig. 12.**

In addition, some results in the paper are confusing.

Fig.3 The ion fragment characteristics of AsA-C ($m/z^- 463$) is very similar to your earlier identified epicatechin-glucuronide conjugate ($m/z^- 463$) in 2016, which makes us puzzled.

Response:

We apologize for the confusion regarding the compound annotation. The MS2 profile of the candidate compound we found in *A. thaliana ans* mutant was similar with that of (-)-epicatechin-5-*O*-glucuronide earlier identified in *Medicago truncatula*. However, the following evidence helps with distinguishing these two compounds. First, the *ans* mutant does not possess the pathway to synthesize flavan-3-ol monomers, meaning that the (-)-epicatechin substrate is absent for further glucuronidation. Second, the absence of a specific ion at m/z 113 (Gu et al., 1999, *Fresenius' Journal of Analytical Chemistry*, 365(6): 553-558) in MS2 profile under negative mode excludes the possibility that the fragment m/z 175 in our candidate is a glucuronic acid moiety. Third, fragment ions m/z 59 and 115 further support that m/z 175 is ascorbate in this paper. To clarify this, we add the above content in the compound deduction part of the revised manuscript. Please see **Lines 144 to 150.**

Line 254. It was described that AsA-[C]_n existed in flowers, berry skins and seeds of grapevine at active stages of PA biosynthesis. So, is there such a possibility that AsA-C was formed during the extraction process, in the process PA cleavage occurred and produced carbocations, which were then trapped by AsA. Combined with your previous papers (“VvLAR1 and VvLAR2 Are Bifunctional Enzymes for Proanthocyanidin Biosynthesis in Grapevine”, 2019), if these conjugates actually exist in grapevine, then in the process of C-type PA polymerization, which of AsA-C and Cys-C played a more important role? It is worth discussing.

Response:

As mentioned in “reply to comment (1)”, the absence of AsA-EC and AsA-C in WT *A. thaliana* extracts reflects the fact that no PA cleavage occurred in our processing method. In the revised manuscript, we have further discussed whether AsA-C plays a more important role than Cys-C in the polymerization of (+)-catechin-type PAs. Please see **Lines 442 to 460** in the revised manuscript. Thank you.

Response to Reviewer #2

Reviewer #2 (Remarks to the Author):

The authors uncover new information on proanthocyanidin (PA) synthesis, specifically identifying the involvement of an ascorbate-catechin conjugate in the polymerisation process and this provides a very significant advance in knowledge. PA is an important plant constituent, so further characterisation of its biosynthesis is of wide interest. Extensive and elegant mass spectrometry analysis and use of an Arabidopsis mutant (*ans4*) enabled identification of the ascorbate-catechin conjugates, and this further enabled identification of the same conjugate in wild type grape. The authors propose an PA polymerisation pathway involving the ascorbate-catechin conjugate.

The conclusions are supported by extensive data, and I have no specific comments/criticisms on this or the methods, which all appear sound and appropriate. However, given that the journal publishes papers of wider interest, I feel that the authors could improve the abstract, introduction and discussion material to improve clarity for the non-expert. I found the text overall to be quite difficult to follow and a clearer introduction to the pathway would assist in understanding the new data. It would be helpful if the extent to which this conjugate is involved in wild type was stated more clearly in the summary and conclusions.

Response:

Thank you for raising this concern to help with improving the manuscript.

We have edited the abstract to simplify the main message by simply posing the question as understanding the PA polymerization mechanism, clarify that the grape is the wild-type, and include reference to the new data in which we infer an *in vivo* role for the new conjugates. To further improve clarity for the non-expert, we have revised the introduction to provide additional information to the pathway products, and emphasize that not all steps in the pathway are conserved. At the same time, we have incorporated a figure (**Fig. 1**) describing PA pathways from Medicago, Arabidopsis and grape in the main text to assist in understanding the known pathway and the new data. In addition, we have edited the discussion to focus more on the source and *in vivo* roles of AsA-[C]_n in both the Arabidopsis *ans* mutant and the wild-type grape.

Although I do not propose that more data are needed for this manuscript, it would be interesting to know to what extent PA polymerisation is influenced by ascorbate concentration (or example in ascorbate deficient (*vtc*) mutants of Arabidopsis) since there is evidence that its concentration is particularly high in developing

siliques (Conklin et al. (2000) *Genetics* 154(2): 847-856). As an aside (which might not be relevant to reproductive tissue), anthocyanin-related genes have lower expression in leaves at high light intensity in *vtc* mutants along with less anthocyanin (Page et al. 2012 *Plant Cell and Environment* 35(2): 388-404) indicating a link between ascorbate synthesis and flavonoid metabolism.

Response:

Thank you for the valuable suggestion. Accordingly, we further measured PA content in the developing siliques of the *vtc2-5 A. thaliana* mutant, which is reported to possess ~70% less AsA than the WT (Becker et al., 2014, *Journal of Experimental Botany*, 65(20): 5903-5918). Levels of both soluble and insoluble PAs in the developing siliques of *vtc2-5* were decreased compared with those of the WT. We also obtained the *vtc2-5:ans-4* double mutant via crossing. By applying thiolysis with excess of Cys to both soluble extracts and insoluble fractions, we found that developing siliques of *vtc2-5:ans-4* contained lower amounts of leucocyanidin-derived carbocation than those of the *ans-4* single mutant. This means that AsA level could regulate the production of leucocyanidin upstream and further affect PA content. We have included these results in **Supplementary Fig. 10 to Supplementary Fig. 12 and the Lines 265 to 284** in the revised manuscript.

Response to Reviewer #3

Reviewer #3 (Remarks to the Author):

What are the noteworthy results? These appear to be note worthy in that this area while an important part of plant metabolism, still seems incompletely known and confusing.

Will the work be of significance to the field and related fields? I believe the work is significant as it establishes an import role of ascorbate utilization in plant metabolism

How does it compare to the established literature? If the work is not original, please provide relevant references.. The work is a significant advance

Does the work support the conclusions and claims, or is additional evidence needed? The work is detailed in as much as it attempts to report

Are there any flaws in the data analysis, interpretation and conclusions? Do these prohibit publication or require revision? Not really, just a bit dense in places

Is the methodology sound? Does the work meet the expected standards in your field? yes

Is there enough detail provided in the methods for the work to be reproduced? Yes

(1) An excellent paper well written English, describing an important alternative pathway for proanthocyanidin (PA) biosynthesis. However the paper is overly complex skirting some issues such as the source of Cys-EC and Cys-C and ascorbyl [C]n. A clearer introduction to the biochemistry and chemistry of PA synthesis after the known enzymatic steps (LAR, ANS, LDOX and ANR) would be helpful, at least to say that the actual pathway is unclear. The authors should refer to papers on the biosynthesis of PAs earlier in the MS. They do include pertinent literature, but the reviewer had to search for them.

Response:

Thank you for the valuable comment. We have addressed this point in the revised manuscript. Our recent papers (refs 15, 16 and 44) describe the complexity of the PA pathway and the fact that it can vary between species. It is really not possible to outline this complexity in any detail in just a few sentences suitable for inclusion in the abstract; that is done (we hope successfully) in the introduction. We have edited the introduction to indicate that not all steps in the pathway are conserved, and provide additional information

to the products in the biochemical pathway. In addition, we think your suggestion in comment (4) is very helpful. Accordingly, we have incorporated a figure (**Fig. 1**) describing PA pathways in three species (Medicago, Arabidopsis and grape) in the main text to help with understanding the source and roles of the known PA starter and extension units.

(2) However the paper clearly identifies the intermediate ascorbyl catechin as important, although it is not clear whether this intermediate is part of the pathway in wild type plants such as Arabidopsis. Also what is the significance or importance of this ascorbyl-C in wt plants apart from grape?

Response:

We have clarified these points, as well as providing additional evidence for the role of the conjugates in PA biosynthesis *in vivo* in wild-type grape (**Fig. 9, Supplemental Fig. 13 and 14, Lines 337 to 368 and Lines 442 to 460** in the revised manuscript). Because the PA pathway is somewhat different in Arabidopsis and grape (and we admit that this pathway is indeed very complex!), we believe that the ascorbate conjugates are not produced in wild-type Arabidopsis (as shown in **Fig. 4c, Fig 5a** and **Fig. 5c**). We believe that extending this work to other plants, although interesting, is beyond the scope of the present manuscript. We are hoping that others working on PA biosynthesis in other species, such as tea and persimmon, will follow up on our findings.

(3) Line 44. The authors mean synthase, not reductase for ANS

Response:

We apologize for this mistake. The “reductase” has been corrected to “synthase” for ANS in the **Line 43**.

(4) The supplementary figure 1a should be included in the main text rather than as a supplementary figure

Response:

Thank you for the suggestion. We have moved the improved PA pathway figure to the main text to help the reader. Please see **Fig. 1** in the revised manuscript.

(5) Line 100 not sure what only refers to

Response:

Sorry for the unclear statement. We changed “in residues only pre-cleaned by chloroform” into “in the samples with chlorophyll removed by chloroform but without acetone/water extraction” to make the

experimental process clear. Please see **Lines 104 to 105**. We have also clarify the relevant information in the corresponding figure legend and the experimental methods.

(6) Fig 1e. statistical analysis should be provided. Explain what the circles mean.

Response:

Statistical analysis has been provided in the revised manuscript with the explanation of the meaning of the circles in the bar plots. Please see revised **Fig 2e**.

(7) Line 155. Rather than around, a range should be indicated

Response:

Thank you. We provide the range of the retention time of AsA-C and AsA-EC in **Line 167**.

(8) Equations in line 241 onwards should be in the methods or supplementary

Response:

The equations for mDP calculation are moved to Methods section. Please see **Line 247, Line 250 and Line 561 onwards**.

(9) Line 286. The word alone is unclear as to what it refers to

Response:

To eliminate the confusion, the sentence was re-written as “there was no oligomeric procyanidin formed in the controls containing (-)-epicatechin, AsA-C or AsA-C-C as single substrates.” Please also see **Lines 318 to 319**.

(10) Line 307. There are three chemicals mentioned, but only starter and extension unit referred to in reference to respectively

Response:

The starter units are the two flavan-3-ol monomers (bracketed to indicate they are to be considered as one class), and the extension units are the carbocations; we have placed a comma after the bracket to clarify this. Please also see **Lines 371 to 372**.

(11) Line 313. Is there a level of detection?

Response:

Yes. The total level of PA extension units produced by the developing siliques of *ans-4* mutant and WT was quantified using butanolysis experiment (**Fig. 2e**). And AsA-C level in *ans-4* was quantified using UHPLC-QqQ against the standard curve of synthesized standard (**Fig. 6b**). AsA-[C]_n does not exist in the WT *A. thaliana* (**Fig. 4 and Fig. 5**).

(12) Line 446. What does cleaned only with chloroform refer to? Unclear

Response:

To avoid the unclear statement, we have re-written the sentence as “and the samples with chlorophyll removed by chloroform but without acetone/water extraction” in **Lines 530 to 531**.

(13) Line 452. Microtube rather than Eppendorf

Response:

Thank you. We changed “Eppendorf tube” into “microtube”. Please also see **Line 534**.

(14) Line 554. While statistical methods are described the results of this analysis do not seem to be reported

Response:

The statistical analysis was reported in **Fig. 2e, Fig. 6b, and Fig. 9** in the main text and **Supplemental Figs. 7c, 10b, 11 and 12** in the Supplemental Information.

(15) Fig 1 and 2. The WT is above the ans mutant in fig 1 but the reverse is true in fig 2. Be consistent

Response:

Thank you for pointing out our mistake. We moved the chromatograms of *ans* mutant above that of the WT in the original Fig. 1 (now **Fig. 2**).

(16) Fig 6. I would be useful to include the tissue type in the figure to make it clear

Response:

Thank you. We added the tissue type along with the developmental stages and chromatograms in the original Fig 6 (now **Fig. 7**) to make it clear.

(17) Fig 7f. Why are there 5 rows of symbols in this figure?

Response:

In Fig. 7f (now **Fig. 8f**), the five rows of substrate symbols contain ten AsA-C or AsA-C-C, and this number just corresponds to the number of extension units involved in PA polymerization in the products. We would like to use Fig 7f (now **Fig. 8f**), on the one hand, to show the types of products formed by the reaction of AsA-C and AsA-C-C with (-)-epicatechin respectively, and on the other hand to help with better understanding the reason why “AsA-C-C + (-)-epicatechin” generated more PAs with odd number of units than the “AsA-C + (-)-epicatechin” combination.

REVIEWER COMMENTS

Reviewer #1 (Remarks to the Author):

I really appreciate that the author has done a lot of solid experimentation. I also believe (catechin)_n: ascorbate conjugates (AsA-[C]_n) do exist in plants. But, I still have some small questions for editors and authors to consider before accepting the article.

The main concerns are as following,

(1) Are AsA-[C]_n conjugates the main constituents in the colored part after butanolysis in Fig. 2d? From Fig. 2e, the total PA can reach 10 mg/g (FW) in the ans mutant, but the free AsA-C in soluble PA is only 0.08 μmol/g (FW) (Fig. 6b), approximately equal to 333 folds ?? Is it indicated that the content of AsA-C in total PA is too low? Can I reasonably infer that leucocyanidin and its other polymers are the main compounds in the colored part in Fig2d? If possible, you can quantify the concentration of leucocyanidin in ans 4 mutant;

(2) From the relative quantitative experimental results of grapes (Fig. 9), it can be found that the content of AsA-C-C and AsA-C-C-C substances is very low, and it is difficult to draw the path of $n > 1$ in Fig. 10. I think the conjugates of catechins, such as AsA-[C], Cys-C, and catechin:methanol conjugate, may be formed in same way. Is there any difference in the nucleophilic strength among these nucleophiles? We know this will make your experiment more difficult, but it will help you to deduce the path of $n > 1$ in Fig. 10.

Reviewer #2 (Remarks to the Author):

I am happy with the revisions made in response to my comments

Reviewer #3 (Remarks to the Author):

The authors have significantly improved their paper and answered queries by this reviewer and by other reviewers

Response to Reviewer #1

Reviewer #1 (Remarks to the Author):

I really appreciate that the author has done a lot of solid experimentation. I also believe (catechin)_n: ascorbate conjugates (AsA-[C]_n) do exist in plants. But, I still have some small questions for editors and authors to consider before accepting the article.

The main concerns are as following,

(1) Are AsA-[C]_n conjugates the main constituents in the colored part after butanolysis in Fig. 2d? From Fig. 2e, the total PA can reach 10 mg/g (FW) in the *ans* mutant, but the free AsA-C in soluble PA is only 0.08 μmol/g (FW) (Fig. 6b), approximately equal to 333 folds ?? Is it indicated that the content of AsA-C in total PA is too low? Can I reasonably infer that leucocyanidin and its other polymers are the main compounds in the colored part in Fig. 2d? If possible, you can quantify the concentration of leucocyanidin in *ans-4* mutant;

Response:

Thank you very much for the valuable questions and the suggestion.

It is reported that leucocyanidin can be converted to colored cyanidin under butanolysis (Smathers and Charley, 1967, *Journal of Food Science*, 32(3): 310-314). To study whether free leucocyanidin contributes a large portion of the red colored products in Fig. 2d, as you suggested, we quantified the concentration of leucocyanidin in *ans-4* mutant 7 DAP siliques as 0.010 μmol g⁻¹ FW, which is close to that of free Cys-C (0.011 μmol g⁻¹ FW) but much lower than that of free AsA-C (0.075 μmol g⁻¹ FW). Compared with total PA or total carbocation (8.58 mg g⁻¹ FW, equal to 14.84 μmol g⁻¹ FW), free leucocyanidin is not the main contributor to the red colored products. This further indicates that the leucocyanidin produced in *Arabidopsis ans-4* mutant are largely consumed to generate polymers or catechin conjugates.

Besides being a soluble conjugate of monomeric catechin, AsA-C can further couple into AsA-[C]_n (n > 1) in the soluble extracts and insoluble fraction of the *ans-4* mutant. Based on the thiolysis results, we calculated the concentration of AsA-C released from soluble polymers as 0.013 μmol g⁻¹ FW and that of AsA-C released from insoluble fraction as 0.043 μmol g⁻¹ FW. Then the concentration ratio between total carbocation (14.84 μmol g⁻¹ FW) and total AsA-C (0.131 μmol g⁻¹ FW) is 113. Thus, the red colored part after butanolysis in Fig. 2d may mainly originate from the catechin extension unit released from the soluble and insoluble fractions. And these catechin extension units can come from either AsA-[C]_n or other potential PA-like molecules. It is reasonable, as you infer, that other unknown polymers could possibly function as carbocation donors under butanolysis. However, we still could not simply underestimate

the role of AsA to initiate very long chain polymers, which could provide a good portion of the carbocations. Using the best mass spectrometry facilities available to us, we were able to trap AsA-[C]_n with a degree of polymerization up to 8 (n = 7). And it is highly possible that AsA-[C]_n with longer chains may exist and eventually become insoluble. Although AsA-[C]_n is so far the most abundant known catechin conjugate, we still expect further work to uncover other PA-like molecules with unconventional starter units to completely resolve the destination of leucocyanidin in the *ans-4* mutant.

In the revised manuscript, for the convenience of comparing concentrations between different compounds, we unified the concentration unit as “ $\mu\text{mol g}^{-1}$ FW” by changing “ mg g^{-1} FW” into “ $\mu\text{mol g}^{-1}$ FW” for total PA quantification in **Fig. 2e**. The data from leucocyanidin measurement in the plant extracts was incorporated as **Supplementary Fig. 1** and the data for the quantification of AsA-C coupled into polymers was added in **Fig 6b**. To maintain logic, the LC/MS chromatograms of the thiolysis of WT soluble extracts in original Fig. 6a were moved to the original Supplementary Fig. 8 (now **Supplementary Fig. 9**); the LC/MS chromatograms of the thiolysis of *ans-4* insoluble fractions in original Supplementary Fig. 8 were moved to the new **Fig. 6a**. The essential words for describing the new data and the corresponding experimental methods were added, and we improved the discussion based on your considerations. Please also see **Lines 113 to 121, Lines 260 to 265, Lines 439 to 444, Lines 596 to 602 and Lines 637 to 647** in the revised manuscript.

(2) From the relative quantitative experimental results of grapes (Fig. 9), it can be found that the content of AsA-C-C and AsA-C-C-C substances is very low, and it is difficult to draw the path of n>1 in Fig. 10. I think the conjugates of catechins, such as AsA-[C], Cys-C, and catechin:methanol conjugate, may be formed in same way. Is there any difference in the nucleophilic strength among these nucleophiles? We know this will make your experiment more difficult, but it will help you to deduce the path of n>1 in Fig. 10.

Response:

Thank you for the comments. It is true that the abundance of AsA-C-C and AsA-C-C-C in the grape berry skins and seeds is much lower than that in 7 DAP siliques of *Arabidopsis ans-4* mutant. And we agree with you that the known conjugates of catechin may be formed in the same way, via nucleophilic attack of AsA/ Cys/ methanol (MeOH) with leucocyanidin-derived carbocation. However, we apologize for not fully understanding your questions: we are not sure how to interpret “in the same way”; as far as the nucleophilicity is concerned, we are not sure if you are questioning AsA to be a reasonable nucleophilic starter unit; because flava-

3-ol:MeOH is not an endogenous conjugate, we are also confused about the particular reason to further include MeOH or catechin:MeOH in the present study. In this response, we provide as much information as we can to hopefully cover the aspects that you may be interested in.

The accumulation of a certain compound *in vivo* depends on its biosynthesis and consumption. Our MS/MS evidence shows AsA-C-C and AsA-C-C-C do exist in grapes and the *in vitro* auto-polymerization experiment suggests that these two compounds potentially provide PA extension units. Unlike Arabidopsis *ans-4* mutant, the grapevine possesses the flavan-3-ol starter units to consume AsA-[C]_n to form PAs. It is also known that during grape berry development, the PA pathway continues switching building block synthesis to polymerization. This means that AsA-C-C and AsA-C-C-C will keep being consumed from pea-sized berries to veraison berries (the samples we studied) until undetectable. By estimating the bond dissociation energy (BDE) with ALFABET tool (St John et al., 2020, *Nature Communications*, 11(1):1-12), we found the C4-O bonds of both AsA-C-C and AsA-C-C-C are slightly weaker than that of AsA-C and much weaker than the C-S bond of Cys-C and the C4-O bond of catechin:MeOH. This indicates that AsA-C-C and AsA-C-C-C more easily donate carbocations than other known catechin conjugates for PA polymerization and may be the reason for the relatively low level of AsA-[C]_n ($n > 1$). As we discussed in the manuscript, the role for AsA-[C]_n in grapes is more like an intermediate to pre-assemble excess leucocyanidin before participating in PA polymerization. Likewise, in the lignin pathway, coniferaldehyde and syringaldehyde are accepted intermediates in monolignol biosynthesis, but their levels are often too low to measure, whereas the corresponding alcohols (the next products) are easy to quantify. Considering this, one may not simply deny the existence of a pathway based on the levels of compounds.

According to classical organic chemistry principles, we believe that two classes of nucleophiles are involved in (catechin)_n:nucleophile ($n > 1$) generation: (1) when trapping leucocyanidin-derived carbocation to form monomeric catechin conjugate, it is required that AsA, Cys or MeOH be deprotonated as the “lone pair” class nucleophile; (2) the further nucleophilic attack for PA-like chain extension requires the last catechin building block to serve as the “ π bond” class nucleophile. When it comes to initiating the carbocation polymerization in plant cells, the dissociation ability of the 3O-H of AsA, S-H of Cys and O-H of MeOH in aqueous environment should be first taken into account. Acid dissociation constant (pK_a) and bond dissociation energy (BDE) can help with understanding the differences as nucleophiles. By referring to previous publications, the pK_a values of the three compounds are as follows: pK_a (AsA, 3O-H) \sim 4.1, pK_a (Cys, S-H) \sim 8.5, pK_a (MeOH, O-H) \sim 15.5 (Kumler and Daniels,

1935, *Journal of the American Chemical Society*, 57(10):1929-1930; Danehy and Parameswaran, 1968, *Journal of Chemical & Engineering Data*, 13(3):386-389; Ballinger and Long, 1960, *Journal of the American Chemical Society*, 82(4):795-798). And the corresponding BDEs are predicted as: BDE (AsA, 3O-H) = 79.5 kcal mol⁻¹, BDE (Cys, S-H) = 85.1 kcal mol⁻¹ and BDE (MeOH, O-H) = 102 kcal mol⁻¹ (St John et al., 2020, *Nature Communications*, 11(1):1-12). Thus, compared with Cys and MeOH, AsA can more readily donate hydrogen ion, making it a “better” nucleophile to trap leucocyanidin *in vivo*. This partially explains the following facts: AsA-C is more abundant than Cys-C *in planta*; although plants produce MeOH (although mainly from pectin in the cell wall), we are not able to trap endogenous catechin:MeOH; epicatechin:MeOH is also not detectable in the plant extracts unless 80% MeOH-based (rather concentrated nucleophile) buffer is used for the extraction as in a recent study (Wang et al., 2020, *The Plant Journal*, 101(1): 18-36). Next, AsA-C and Cys-C are the two potential “ π bond” nucleophiles to further attack leucocyanidin-derived carbocation for PA-like chain elongation *in planta*. And the nucleophilicity at the C8 position of the two conjugates is mainly determined by the electron densities of the atom or the atomic charges in the present case. By using ACC II web server (Raček T et al., 2020, *Nucleic Acids Research*, 48(W1): W591-W596), the atomic charges of the C8 atom in AsA-C and Cys-C were predicted as -0.3070 and -0.2871 respectively. This means that the C8 position in AsA-C is more electron rich and thus with a stronger nucleophilicity compared with that in Cys-C, further supporting the fact that Cys-C-C or Cys-C-C-C were not detected in the plants. Furthermore, we compared the predicted C8 atomic charges among AsA-C, AsA-C-C, catechin and procyanidin B3 as follows: catechin (-0.3094) < AsA-C (-0.3070) < AsA-C-C (-0.3065) < procyanidin B3 (-0.2828). This indicates that nucleophilicity of AsA-C and AsA-C-C are potentially similar with catechin and stronger than procyanidin dimer and might further suggest that the polymerization processes of AsA and PA share commonality. Because of space, we have not presented all these arguments in the revised manuscript, but hope we have now included enough on bond strengths and dissociation energies to address your question.

In addition, the subcellular compartment of the nucleophiles is also important for catechin conjugate formation. We have shown that AsA-EC does not exist in the wild-type (WT) Arabidopsis and grape, although previous studies suggest Arabidopsis WT plants do possess epicatechin carbocation (Wang et al., 2020, *The Plant Journal*, 101(1): 18-36; Jun et al., 2021, *Science Advances*, 7(20): eabg4682). This indicates that AsA subcellular localization is strictly restricted. In the discussion part of the manuscript, we have also emphasized the point that the use of AsA as a co-substrate for the enzyme that makes leucocyanidin might ensure high local

concentrations of AsA, which means that so far AsA is the most reasonable nucleophile to co-localize with leucocyanidin and sequentially trap leucocyanidin-derived carbocation to form AsA-[C]_n.

We hope the above response will successfully address the potential points you concerned. Meanwhile, we think some of the content stated here is helpful in understanding the results and we have further added a supplementary table for BDE prediction (**Supplementary Table 1**) and the relative results and discussion in the revised manuscript. Please also see **Lines 317 to 321**, **Lines 414 to 420**, **Lines 475 to 482** and **Lines 741 to 744** in the revised main text.

Response to Reviewer #2

Reviewer #2 (Remarks to the Author):

I am happy with the revisions made in response to my comments.

Response:

Thank you again for the effort to help with improving our work.

Response to Reviewer #3

Reviewer #2 (Remarks to the Author):

The authors have significantly improved their paper and answered queries by this reviewer and by other reviewers

Response:

We really appreciate your queries and suggestions for improving the manuscript.